# Service availability and readiness for basic emergency obstetric and newborn care: Analysis from Nepal Health Facility Survey 2021

Achyut Raj Pandey[1]*, Bikram Adhikari[1], Bipul Lamichhane[1], Deepak Joshi[1], Shophika Regmi[1], Bibek Kumar Lal[2], Sagar Dahal[2], Sushil Chandra Baral[1]

1 HERD International, Lalitpur, Nepal, 2 Family Welfare Division, Department of Health Services, Ministry of Health and Population, Kathmandu, Nepal

* achyutrajpandey2014@gmail.com

## Abstract

### Background

Although there has been a significant focus on improving maternal and newborn health and expanding services in Nepal, the expected positive impact on the health of mothers and newborns has not been achieved to the desired extent. Nepal not only needs to focus on improving access to and coverage of services but also the quality to achieve Sustainable Development Goals (SDG) by 2030. In this context, we aimed to analyze Basic Emergency Obstetric and Neonatal Care (BEmONC) service availability and readiness in Health Facilities (HFs) of Nepal.

### Methods

We analyzed data from nationally representative Nepal Health Facility Survey (NHFS), 2021. BEmONC service availability and readiness in HFs was measured based on the "Service Availability and Readiness" manual of World Health Organization (WHO). We measured service availability by seven BEmONC signal functions. The readiness score was calculated for three domains- guidelines and staff training, essential equipment/supplies, and essential medicines on a scale of 100, and the average score for the three domains was the overall readiness score. We performed weighted descriptive and inferential analysis to account complex survey design of NHFS 2021. We summarized continuous variables with descriptive statistics like mean, standard deviation, median and interquartile range whereas categorical variables with percent and 95% confidence interval (CI). We applied simple, and multivariate linear regression to determine factors associated with the readiness of HFs for BEmONC services, and results were presented as beta (β) coefficients and 95% CI.

### Results

Of total 804 HFs offering normal vaginal delivery services, 3.1%, 89.2%, 7.7% were federal/provincial hospitals, local HFs, and private hospitals respectively. A total of 45.0% (95% CI:

**Data Availability Statement:** https://dhsprogram.com/data/available-datasets.cfm.

**Funding:** The author(s) received no specific funding for this work.

**Competing interests:** The authors have declared that no competing interests exist.

34.9, 55.6) federal/provincial hospitals, 0.3% (95% CI: 0.2, 0.6), local HFs (district hospital, primary health care centers, health posts, urban health centers) and 10.5% (95% CI: 6.6, 16.4) private hospitals, had all seven BEmONC signal functions. The overall readiness of federal/provincial hospitals, local HFs, and private hospitals were 72.9±13.6, 54.2±12.8, 53.1±15.1 respectively. In multivariate linear regression, local HFs (β = -12.64, 95% CI: -18.31, -6.96) and private hospitals had lower readiness score (β = -18.08, 95% CI: -24.08, -12.08) compared to federal/provincial level hospitals. HFs in rural settings (β = 2.60, 95% CI: 0.62, 4.58), mountain belts (β = 4.18, 95% CI: 1.65, 6.71), and HFs with external supervision (β = 2.99, 95% CI:1.08, 4.89), and quality assurance activities (β = 3.59, 95% CI:1.64, 5.54) had better readiness scores.

## Conclusion

The availability of all seven BEmONC signal functions and readiness of HFs for BEmONC services are relatively low in local HFs and private hospitals. Accelerating capacity development through training centers at the federal/provincial level, onsite coaching, and mentoring, improving procurement and supply of medicines through federal/provincial logistic management centers, and regular supportive supervision could improve the BEmONC service availability and readiness in facilities across the country.

## Background

As a part of Sustainable Development Goal (SDG) 3.1, Nepal is committed to reduce maternal mortality to less than 70 per 100,000 live births [1, 2]. While the annualized rate of change in maternal mortality rate was -2.9 between 1990 to 2017, it has remained -0.9 between 2005 to 2017 indicating a slower decline in recent decade [3]. If this current rate of decline continues, Nepal is at risk of failing to meet the SDG's target. Similarly, Neonatal Mortality Rate (NMR) in Nepal fell from 58 deaths per 1000 live births in 1990 to 21 deaths per 1000 live births in 2022 [4, 5]. Although notable progress has been made in reducing neonatal mortality in general, the progress has been relatively slow in recent years. For example, NMR has remained stagnant at 12 per 1000 live birth between 2016 and 2022 [5] which challenges Nepal's ability to achieve the SDG target of 12 or fewer newborn deaths per 1000 live births by 2030 [6].

Timely access to emergency obstetric and newborn care (EmONC) services is one of the key drivers to reduce maternal and neonatal mortality globally [7, 8]. A study by Bhutta et al. in 2014 showed that around 150,000 maternal deaths, 550,000 stillbirths and 790,000 neonatal deaths could be prevented globally by scaling up care during labor and childbirth including complications [9]. Similarly, studies have estimated that approximately 15–20% of all pregnancies are complicated and require hospital attention [10, 11] thus requiring health facilities (HFs) providing delivery services to deal with the complicated cases in-order to handle such cases as they arise. According to a previous systematic review and Delphi estimation, skilled childbirth care alone would prevent 25% of intrapartum-related neonatal deaths [12]. Similarly, Basic and comprehensive emergency obstetric care was estimated to prevent 40% and 85%neonatal deaths due to intrapartum events respectively [9, 12].

With the objective of reducing preventable maternal mortality, in 2020, the World Health Organization (WHO) and United Nations Population Fund (UNFPA) set a global target of ensuring that at least 60% of the population can physically reach the nearest EmONC facility within two-hours travel time. Moreover, the objective was established to ensure that 80% of

countries have a minimum of 50% of their population capable of reaching the closest emergency obstetric and newborn health facility within two-hours travel time [7]. The access to EmONC level facilities is also a critical intervention for reducing neonatal mortality and preventing stillbirths and has also been a critical component of Every Newborn Action Plan (ENAP) [13].

In Nepal, the Safe Motherhood and Newborn Health (SMNH) Road Map 2030, recommends encouraging all women to give birth at a Basic Emergency Obstetric and Newborn Care (BEmONC) or Comprehensive Emergency Obstetric and Newborn Care (CEmONC) facility if such facilities are within 2 hours walking distance from their house [14]. In line with the global recommendations, SMNH roadmap also recommends that the met need of Emergency Obstetric Care (EOC)should be 100% [14]. However, the latest routine data from the Department of Health Services (DoHS) revealed that the nationwide availability of met needs of EOC hovered around 11–12% for three fiscal years from 2017/18 to 2019/20, then dropped to roughly 8% in 2020/21. These figures on met need of EOC are concerning and signals reduction in utilization or delivery of complication treatment during pregnancy [15]. Data show variation across the seven provinces with met need of EOC ranging from 3.5% to 10.5% with highest in Lumbini province (10.5%) and lowest in Madhesh (3.5%) [15].

A rigorous assessment of the extent to which facilities offering delivery services are prepared to provide emergency obstetric care becomes crucial. In this context, we aimed to determine the BEmONC service availability and readiness of Health facilities offering normal vaginal delivery services and determine factors associated with BEmONC service readiness using data from Nepal Health Facility Survey (NHFS) 2021 [16].

## Methods

We performed secondary data analysis of a nationally representative data from NHFS 2021 [16]. The NHFS 2021 was implemented by New ERA, a research firm, with the support from Ministry of Health and Population and ICF International. Since it is the only data source with the nationally representative assessment of availability of services and readiness of facilities to provide BEmONC services, this data source is used in this study.

### Study setting

Nepal falls among low and middle income countries with a human development index of 0.587 [17]. According to the recent housing and population census of 2021, population of Nepal has reached 29.2 million of which 6.07% reside in mountain region, 40.31% reside in hill region and 53.61% reside in Terai region. Nepal has seven provinces, 77 districts and 753 local governments [18].

**Health care delivery system in Nepal.** The overall health system in Nepal is guided by its constitution that guarantees free health care services including Maternal Newborn and Child Health Services (MNCH) to its citizen. Nepal has federalized health system in place to cater the health needs of people through three tiers of governments: federal, provincial, and local governments. At the local level, the health system consists of basic hospitals, Primary Health Care Centers (PHCCs), Health Posts (HPs), Basic Health Service Centers (BHSCs), Urban Health Clinics (UHCs), Community Health Units (CHUs), as well as outreach clinics at community level. While birthing service is available at HP level facilities, basic hospitals are expected to offer BEmONC services [14, 19]. Hospitals at district level and higher level facilities are supposed to offer CEmONC [14, 15, 20] as per the current structure. The provincial and federal health system includes central level hospitals providing comprehensive emergency

obstetric, neonatal care, specialist health services and super specialized maternal health related services [20, 21].

## Sample and sampling techniques

The NHFS 2021 collected data from government-managed facilities, private not-for-profit organizations, NGOs, for-profit organizations, and mission/faith organizations across all 77 districts. In NHFS 2021, 1,633 HFs were chosen from a master list of 5,681 eligible HFs at random using equal probability systematic sampling, of which 7 HFs were excluded due to duplication. Of the total effective sample size of 1,626 HFs, surveys were successfully completed in 1,576 HFs [16]. The sampling approach NHFS 2021 is briefly presented in Fig 1. A total of 786 HFs from NHFS 2021 HFs offering normal delivery services (excluding HTCs) were used for further analysis (Fig 1). Sampling process is described in detail elsewhere [16].

## Data collection

The data collection of NHFS 2021 was performed between 27 January 2021, and 28 September 2021. The data collection process had to be halted for three months between May to July due to COVID-19-imposed lockdowns in Nepal [16].

The NHFS 2021 used four main tools for data collection, namely a) facility inventory questionnaire, b) healthcare worker interview questionnaire, c) observational protocol (for Antenatal Care (ANC), family planning, sick child services, labor, and childbirth) and d) exit interview questionnaires (for ANC and family planning clients and for caretakers of sick children). For this article, we analyzed variables from the facility inventory questionnaire and one variable on staff training from the health worker interview questionnaire [16].

## Outcome variables and measurement approach

The variables for services availability and readiness of HFs for BEmONC were selected based on the WHO service availability and readiness assessment (SARA) manual [22].

**Services availability.**   The service availability of HFs for BEmONC was measured based on seven BEmONC signal functions of delivery and newborn care services which included parenteral antibiotics, parenteral oxytocins, parenteral anticonvulsant, assisted vaginal delivery, manual removal of placenta, removal of retained product of conception, and neonatal resuscitation. The seven signal functions measure the capacity of HFs to treat obstetric and newborn emergencies [3]. The health facility was defined to have availability of BEmONC signal function of delivery and newborn care if the facilities reported applying or carrying them out within the past 3 months preceding the survey. The availability of BEmONC signal functions was measured based on the reporting by the facility staff members and observation by the enumerators during facility visits.

**Services readiness.**   The service readiness of HFs for BEmONC was measured based on the availability and functioning of tracer items categorized into three domains- guidelines and staff training (2 items), essential equipment and supplies (14 items) and essential medicine and commodities (11 items). The list of tracer items of each domain is selected based on previous studies [19] and availability of information in the NHFS dataset. Detailed information about tracer items and score calculation process is present in the S1 Table. Readiness score of HFs to provide BEOmNC services was calculated using WHO's SARA manual [22]. The items in each domain were recorded as binary variables, taking 1 for availability and 0 for absence of the item in the facility. The mean score for each domain was computed by adding items that were divided by the number of items and multiplied by 100. The average score from the three domains was the readiness score. Each domain contributes to the overall score by 33.3%.

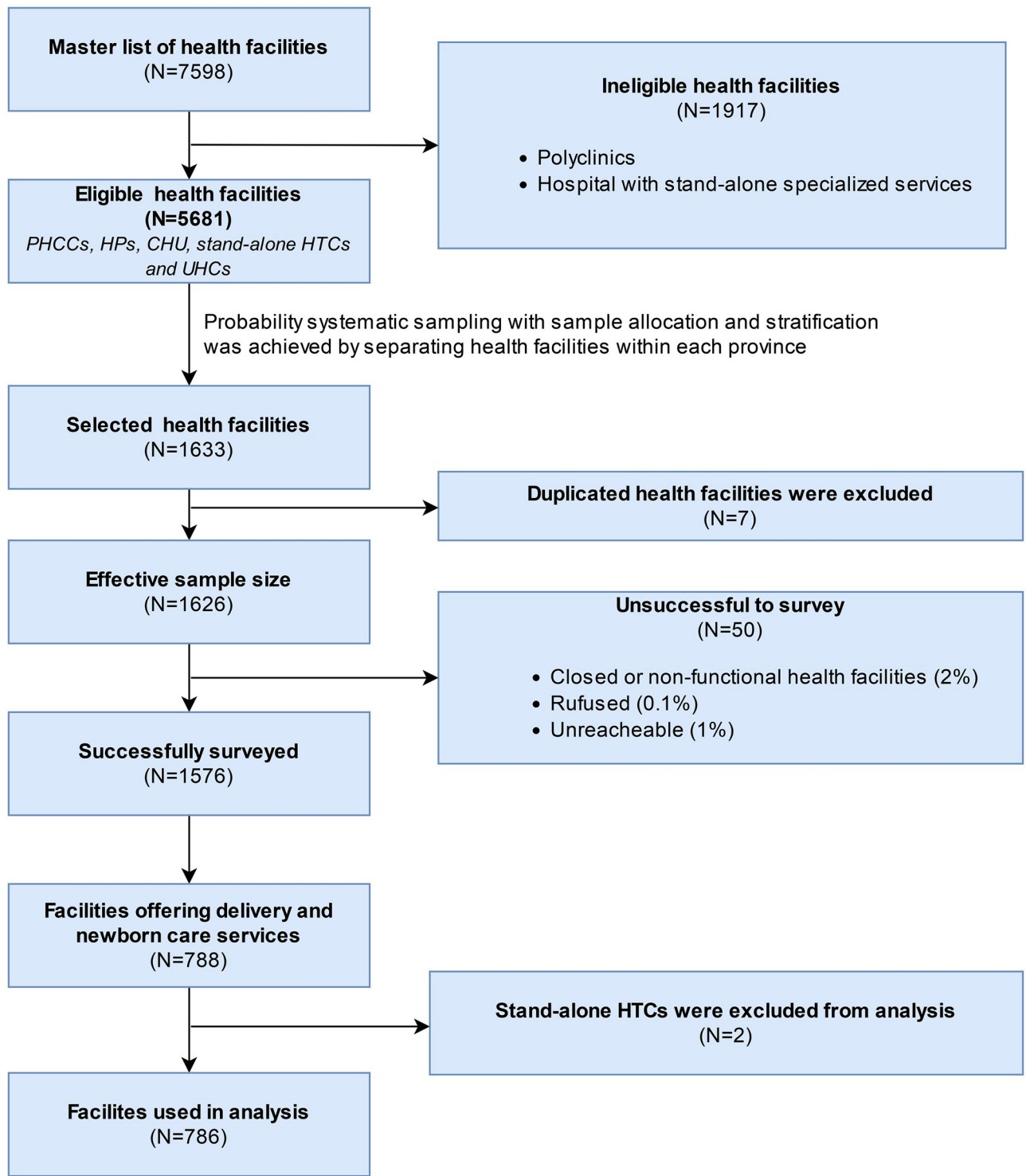

**Fig 1. Flowchart showing sample selection (NHFS 2021).**

### Independent variables

The independent variables included ecological region (hill, mountain, terai), location of facility (rural, urban), province (Koshi, Madhesh, Bagmati, Gandaki, Lumbini, Karnali, Sudurpaschim), type of facility (federal / provincial hospital, local HFs, private hospital), external supervision in past four months (present, absent), duty schedule or call list for 24-hour staff assignment (yes, no), reviews of maternal or newborn deaths (reviewed, not reviewed), system of determining and reviewing clients' opinion (yes, no), quality assurance activities (performed, not performed), frequency of health facility administrative and management meeting (none, sometimes, monthly), number of delivery beds and number of total health workers. Independent variables are described in detail in S2 Table.

### Statistical analysis

We used R version 4.2.0 [23] for statistical analysis and performed weighted analysis using "survey" [24] package to account complex survey design of NHFS 2021. Continuous variables were summarized with mean, standard deviation (SD), median and interquartile range (IQR) whereas categorical variables were summarized with weighted percent and 95 percent confidence interval (CI). Univariate and multivariate weighted linear regression analyses were conducted to assess the association of readiness of HFs to BEmONC and independent variables. These independent variables included ecological region, location of facility, province, type of facility, presence of external supervision in the past four months, duty schedule or call list for 24-hour staff assignment, reviews of maternal or newborn deaths, system of determining and reviewing clients' opinion, quality assurance activities, frequency of health facility meeting, number of delivery beds and number of total health workforce. We used variance inflation factor (VIF) to check for multi-collinearity among independent variables. A p-value of less than 0.05 was considered statistically significant.

In addition, we plotted map in QGIS 3.22.7-Białowieża [18] from freely available GPS dataset of the HFs from https://dhsprogram.com/data/available-datasets.cfm [18]. Publicly available district wise shape file with EPSG:4326 projection system was taken official website of Survey Department of Ministry of Land Management, Cooperatives and Poverty Alleviation, Government of Nepal [25].

**Ethical approval.** The study involves secondary analysis of data from NHFS 2021; hence no ethical approval was sought for this study. In the original NHFS 2021 survey, ethical clearance was obtained from the Ethical Review Board of Nepal Health Research Council and ICF International [16]. In the survey, enumerators provided details on risk and benefits of the study and voluntariness of the participation. Written informed consent was obtained from study participants (health facility head, service providers, and clients) before enrolling them in the study.

## Results

Table 1 presents the characteristics of HFs offering delivery and newborn care services stratified by level of facility. Of the total facilities offering delivery and newborn care services, 3.1% (95% CI: 2.5, 3.9) facilities were federal/provincial hospital, 89.2% (95%CI: 87.2, 91.0) were local HFs and 7.7% (95% CI: 6.1, 9.5) were private hospitals. Majority of HFs were from hilly region(61.4%; 95% CI: 56.3, 66.2). In terms of province, most HFs were from Bagmati (18.8%; 95% CI: 14.9, 23.5), followed by Lumbini (16.9%; 95%CI: 13.3, 21.3), Koshi (16.7%; 95% CI: 13.0, 21.2) and Sudurpaschim (16.1%; 95% CI: 13.1, 19.8). Furthermore, external supervision was present in 68.1% (95% CI: 63.2, 72.7) HFs, quality assurance activities were performed in 31.0% (95% CI: 26.4, 36.1) HFs, review of maternal and newborn deaths was done in 32.7%

**Table 1. Characteristics of facilities offering delivery and newborn care services.**

| Characteristic | n | Overall n = 804 % (95% CI) | Federal/provincial hospitals, n = 25 % (95% CI) | Local HFs n = 718 % (95% CI) | Private hospital n = 61 % (95% CI) |
|---|---|---|---|---|---|
| **Ecological region** | | | | | |
| Hill | 493 | 61.4 (56.3, 66.2) | 51.6 (41.3, 61.8) | 63.3 (57.7, 68.6) | 42.8 (32.5, 53.8) |
| Mountain | 136 | 17.0 (13.5, 21.1) | 16.7 (10.3, 26.0) | 18.2 (14.4, 22.8) | 2.1 (0.8, 5.8) |
| Terai | 174 | 21.7 (17.8, 26.1) | 31.7 (22.8, 42.0) | 18.5 (14.3, 23.5) | 55.0 (44.3, 65.4) |
| **Province** | | | | | |
| Koshi | 134 | 16.7 (13.0, 21.2) | 17.9 (11.2, 27.2) | 16.1 (12.0, 21.2) | 23.1 (16.5, 31.4) |
| Madhesh | 61 | 7.6 (5.2, 11.1) | 11.2 (6.1, 19.5) | 6.7 (4.1, 10.8) | 16.8 (11.4, 23.9) |
| Bagmati | 151 | 18.8 (14.9, 23.5) | 16.7 (10.3, 26.0) | 17.2 (13.1, 22.4) | 38.3 (27.6, 50.2) |
| Gandaki | 92 | 11.4 (8.6, 15.0) | 13.4 (7.8, 22.1) | 11.7 (8.6, 15.8) | 6.8 (3.9, 11.5) |
| Lumbini | 136 | 16.9 (13.3, 21.3) | 15.2 (9.1, 24.4) | 17.4 (13.4, 22.3) | 11.4 (7.2, 17.6) |
| Karnali | 100 | 12.4 (9.8, 15.7) | 12.3 (6.9, 20.8) | 13.4 (10.5, 17.1) | 0.9 (0.2, 3.6) |
| Sudurpaschim | 130 | 16.1 (13.1, 19.8) | 13.4 (7.8, 22.1) | 17.4 (14.0, 21.5) | 2.7 (1.2, 6.1) |
| **Location** | | | | | |
| Urban | 344 | 42.8 (37.9, 47.9) | 95.5 (88.7, 98.3) | 36.5 (31.2, 42.1) | 95.0 (90.4, 97.5) |
| Rural | 460 | 57.2 (52.1, 62.1) | 4.5 (1.7, 11.3) | 63.5 (57.9, 68.8) | 5.0 (2.5, 9.6) |
| **External supervision** | | | | | |
| No | 256 | 31.9 (27.3, 36.8) | 26.8 (18.7, 37.0) | 30.9 (25.9, 36.4) | 45.3 (35.3, 55.7) |
| Yes | 548 | 68.1 (63.2, 72.7) | 73.2 (63.0, 81.3) | 69.1 (63.6, 74.1) | 54.7 (44.3, 64.7) |
| **Duty scheduled in 24 hours** | | | | | |
| No | 629 | 78.2 (74.4, 81.6) | 8.9 (4.5, 16.7) | 85.3 (81.1, 88.6) | 24.3 (15.0, 36.8) |
| Yes | 175 | 21.8 (18.4, 25.6) | 91.1 (83.1, 95.5) | 14.8 (11.4, 18.9) | 75.7 (63.2, 85.0) |
| **Quality assurance activities** | | | | | |
| Not Performed | 555 | 69.0 (63.9, 73.6) | 52.8 (42.4, 62.9) | 69.4 (63.7, 74.4) | 71.4 (60.6, 80.3) |
| Performed | 249 | 31.0 (26.4, 36.1) | 47.2 (37.1, 57.6) | 30.7 (25.6, 36.3) | 28.6 (19.7, 39.4) |
| **Review maternal and newborn deaths** | | | | | |
| No | 542 | 67.3 (62.4, 71.9) | 26.9 (18.7, 37.0) | 70.3 (64.9, 75.3) | 48.9 (38.7, 59.2) |
| Yes | 263 | 32.7 (28.1, 37.6) | 73.1 (63.0, 81.3) | 29.7 (24.7, 35.1) | 51.1 (40.8, 61.3) |
| **System for review of client's opinion** | | | | | |
| No | 761 | 94.7 (92.2, 96.4) | 84.2 (75.1, 90.4) | 96.1 (93.2, 97.78) | 82.1 (72.6, 88.8) |
| Yes | 43 | 5.3 (3.6, 7.8) | 15.8 (9.6, 24.9) | 3.90 (2.22, 6.78) | 17.9 (11.2, 27.4) |
| **Total health workforce,** *Mean±SD; median (Q1, Q3)* | 804 | 27.2±97.1; 7.0 (6.0, 11.0) | 158.1±248.3; 72.7 (52.2, 133.5) | 8.6±6; 7. (6.0, 9.) | 191.6±245.1; 79.0 (38.3, 205.5) |
| **Number of delivery beds,** *Mean±SD; median (Q1, Q3)* | 804 | 1.3±1.1; 1.0 (1.0, 1.0) | 2.6±3.8; 2.0 (2.0, 2.0) | 1.2±0.4; 1.0 (1.0, 1.0) | 2.1±2.4; 2.0 (1.0, 2.0) |
| **Total facilities** | 804 | 100.0 | 3.1 (2.5, 3.9) | 89.2 (87.2, 91.0) | 7.7 (6.1, 9.5) |

n: frequency; %: percent; SD: standard deviation; Q1: first quartile, Q3: third quartile; CI: confidence interval

All percents and column are column percent

(95% CI: 28.1, 37.6) HFs, 24-hour duty schedule was present in about one quarter (21.8%; 95% CI: 18.4, 25.6) of the HFs, and client opinion was reviewed in 5.3% (95% CI: 3.6, 7.8) HFs. The median delivery bed in federal/provincial, local HFs and private hospitals were 2.0, 1.0, and 2.0 respectively. The median health workforce in federal/provincial hospitals, local HFs and private hospitals were 72.7, 7.0, and 79.0 respectively.

Fig 2 presents the availability of seven BEmONC signal functions among facilities offering delivery and newborn care services. In 2021, majority HFs (88.2%) had parenteral oxytocics,

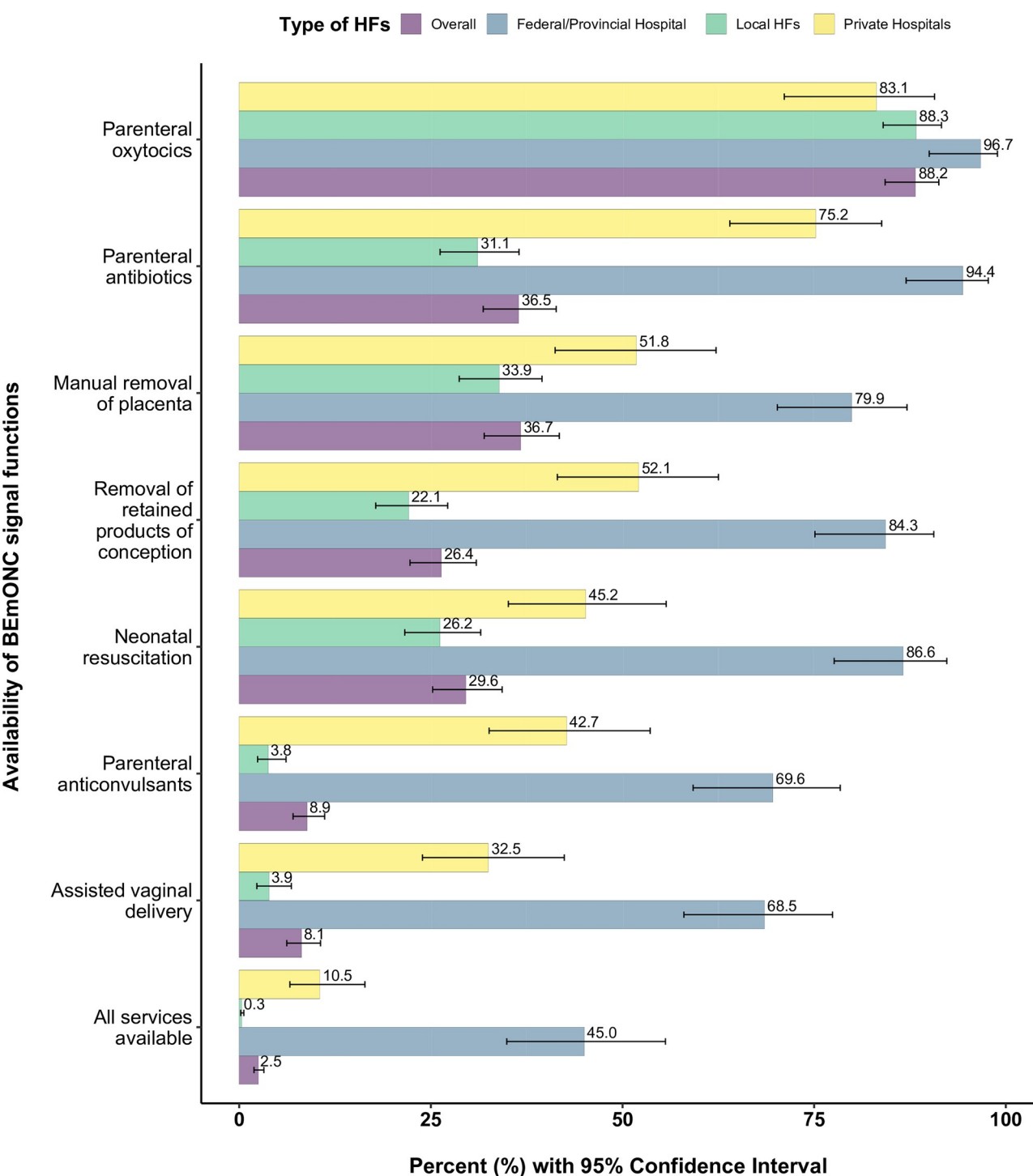

**Fig 2. Availability of BEmONC signal functions by level of facility.**

and only 8.1% HFs (95% CI: 6.2, 10.6) had assisted vaginal delivery. A total of 45.0% (95% CI: 34.9, 55.6) federal/provincial hospitals, 0.3% (95% CI: 0.2, 0.6) of local HFs and 10.5% (95% CI: 6.6, 16.4) of private hospitals with 2.5% (95% CI: 1.9, 3.2) of facilities at national level had all seven BEmONC signal functions.

The availability of all BEmONC signal functions was highest in Koshi (3.7%; 95% CI: 2.1, 6.5) followed by Lumbini (2.8%; 95% CI: 1.5, 5.0) and Bagmati Province (2.2%; 95% CI: 1.2, 4.1). The availability of all BEmONC services was lower in the facilities from rural area (0.06%; 95% CI: 0.01, 0.44) as compared to facilities from urban areas (5.7%; 95% CI: 4.4, 7.5) and in terai belt (6.4%; 95% CI: 4.4, 9.3) compared to hilly region (1.6%; 95% CI: 1.1, 2.4) and mountain region (0.8%; 95% CI: 0.3, 2.3) [S3 Table].

Table 2 presents the availability of tracer items in different types of HFs in NHFS 2021. The availability of guidelines for service delivery was highest in federal/provincial hospitals (22.5%,

**Table 2. Availability of tracer items by level of facility.**

| Characteristic | Tracer items | Overall, n = 804 % (95% CI) | Federal/Provincial hospitals, n = 25 % (95% CI) | Local HFs, n = 718 % (95% CI) | Private hospitals, n = 61 % (95% CI) |
|---|---|---|---|---|---|
| Guidelines and staff training | Guidelines * | 12.8 (9.6, 16.7) | 22.5 (15.0, 32.3) | 12.9 (9.5, 17.3) | 7.4 (4.3, 12.5) |
| | Staff training # | 29.6 (25.1, 34.5) | 58.4 (47.9, 68.2) | 30.5 (25.5, 35.9) | 7.0 (4.1, 11.9) |
| Essential equipment and supplies | Emergency transport** | 81.5 (77.1, 85.2) | 95.5 (88.5, 98.3) | 79.8 (74.9, 83.9) | 95.8 (91.2, 98.1) |
| | Sterilization | 66.1 (61.1, 70.9) | 91.1 (83.1, 95.5) | 62.7 (57.1, 67.9) | 96.8 (92.3, 98.7) |
| | Exam light | 93.8 (90.9, 95.8) | 97.8 (91.5, 99.4) | 93.7 (90.5, 95.9) | 93.7 (81.5, 98.1) |
| | Delivery package ‡ | 97.7 (95.8, 98.8) | 98.9 (92.5, 99.8) | 98.4 (96.4, 99.3) | 89.9 (75.5, 96.3) |
| | Suction apparatus | 65.7 (60.6, 70.4) | 100.0 (100.0, 100.0) | 62.5 (56.9, 67.8) | 89.1 (78.7, 94.7) |
| | Manual vacuum extractor | 23.2 (19.4, 27.4) | 88.8 (80.5, 93.9) | 17.9 (14.1, 22.6) | 57.9 (47.4, 67.8) |
| | Vacuum aspirator | 20.9 (17.5, 24.8) | 94.4 (87.3, 97.7) | 14.4 (11.0, 18.6) | 66.8 (56.1, 76.1) |
| | Resuscitation bag and mask | 91.6 (88.7, 94.0) | 98.9 (92.5, 99.8) | 92.1 (88.7, 94.5) | 83.8 (73.9, 90.4) |
| | Delivery bed | 98.7 (97.0, 99.5) | 98.8 (92.1, 99.8) | 99.1 (97.2, 99.7) | 94.7 (81.3, 98.6) |
| | Blank partograph | 90.4 (87.4, 92.7) | 97.8 (91.5, 99.4) | 91.9 (88.6, 94.3) | 69.9 (59.9, 78.2) |
| | Gloves | 97.5 (95.3, 98.6) | 97.8 (91.5, 99.4) | 97.8 (95.3, 99.0) | 93.1 (81.4, 97.7) |
| | Infant weighting machine | 78.7 (74.2, 82.5) | 63.0 (52.5, 72.4) | 80.6 (75.6, 84.8) | 62.3 (51.6, 71.9) |
| | BP apparatus | 95.3 (92.4, 97.1) | 97.8 (91.5, 99.4) | 95.5 (92.3, 97.5) | 91.1 (80.2, 96.3) |
| | Soap water | 97.4 (95.2, 98.6) | 98.9 (92.5, 99.8) | 97.6 (95.2, 98.8) | 94.5 (81.3, 98.5) |
| Essential medicines | Injectable antibiotics | 66.1 (61.0, 70.8) | 94.4 (87.3, 97.7) | 64.3 (58.6, 69.5) | 75.7 (66.1, 83.4) |
| | Injectable uterotonic | 97.0 (94.9, 98.3) | 97.8 (91.5, 99.4) | 98.0 (95.6, 99.1) | 85.4 (72.9, 92.7) |
| | mGSO4 injection | 70.7 (65.6, 75.2) | 95.5 (88.7, 98.3) | 69.9 (64.3, 74.9) | 69.7 (58.8, 78.8) |
| | Diazepam | 28.2 (24.0, 32.7) | 87.7 (79.2, 93.1) | 22.4 (18.1, 27.4) | 70.7 (60.8, 78.9) |
| | IV set | 97.2 (95.4, 98.3) | 98.9 (92.5, 99.8) | 97.8 (95.9, 98.8) | 89.4 (79.1, 94.9) |
| | Skin disinfectant | 82.4 (78.4, 85.7) | 87.7 (79.2, 93.1) | 83.8 (79.4, 87.4) | 63.7 (53.7, 72.6) |
| | Eye ointment | 7.8 (5.5, 10.8) | 20.2 (13.1, 29.8) | 6.9 (4.5, 10.4) | 12.7 (8.3, 19.0) |
| | Chlorhexidine | 80.2 (75.9, 83.8) | 85.4 (76.5, 91.4) | 81.5 (76.8, 85.5) | 61.9 (51.8, 71.0) |
| | Ceftriaxone | 38.1 (33.5, 43.0) | 92.1 (84.3, 96.2) | 32.0 (27.1, 37.4) | 87.4 (81.0, 91.9) |
| | Amoxicillin | 62.2 (57.1, 67.1) | 77.4 (67.5, 84.9) | 62.3 (56.7, 67.7) | 54.3 (43.7, 64.5) |
| | Gentamicin | 79.8 (75.2, 83.7) | 86.4 (77.6, 92.1) | 80.0 (74.8, 84.3) | 75.0 (66.8, 81.8) |

%: percent; CI: Confidence Interval

#: Facility having at least one service provider trained on delivery services (skill births attendant (SBA), advanced SBA, maternal and newborn health update, routine care during labor and normal vaginal delivery, active management of third stage labor (AMTSL)) in last 24 months preceding the survey

* The HFs in which enumerator observed Nepal Medical Standards (NMS) Volume III, Reproductive Health (RH) clinical protocols, or any other clinical protocols/guidelines were considered to have guidelines

** Facility had a functioning ambulance or other vehicle for emergency transport stationed at the facility and had fuel available on the day of the survey, or facility has access to an ambulance or other vehicle for emergency transport that is stationed at another facility or that operates from another facility.)

‡ Either the facility had a sterile delivery pack available at the delivery site or else all of the following individual equipment was present: cord clamp, episiotomy scissors, scissors (or blade) to cut cord, suture material with needle, and needle holder and four-piece wrapper

95% CI: 15.0, 32.3) and least in private hospitals (7.4%; 95% CI: 4.3, 12.5). Around 29.6% (95% CI: 25.1, 34.5) of all facilities, 58.4% (95% CI: 47.9, 68.2) of federal/provincial hospitals, 30.5% (95% CI: 25.5, 35.9) of local HFs and 7.0% (95% CI: 4.1, 11.9) of private hospitals had at least one service provider trained on delivery services in past 24 months preceding the survey. Similarly, the proportion of facilities having at least one dedicated bed for safe delivery was 98.7% (95% CI: 97.0, 99.5). Furthermore, more than 95% of all HFs had gloves, soap and water, and BP apparatus.

The data presented in Fig 3 highlights a significant disparity in the BEmONC service readiness score across different districts and types of HFs. The graph indicates that some areas and certain types of facilities may be better equipped to provide BEmONC services than others.

Table 3 presents the comparison of overall and domain wise readiness score by type of facility. The aggregated weighted BEmONC service readiness score was 54.7±13.4 and domain wise average score for guideline and staffs, equipment and supplies and essential medicine and commodities domains were 21.2±28.2, 78.5±12.3 and 64.5±16.8 respectively. The mean BEmONC readiness score independent of delivery load for federal/provincial hospital, local HFs, and private hospitals were 72.9±13.6, 54.2±12.8, 53.1±15.1 respectively.

The guidelines and staff training domain contributed least to the overall readiness score for each type of health facility. The median score for guidelines and staff training was 50 (Q1 & Q3: 0.0, 50.0) for federal/provincial hospital whereas 0.0 (Q1 & Q3: 0.0, 50.0) for local HFs and 0.0 (Q1 & Q3: 0.0, 0.0) for private hospitals. Whereas the essential equipment and supplies, and essential medicine domains contributed more to the overall readiness score for each facility type.

Factors associated with BEmONC service readiness are presented in Table 4. In bivariate analysis, ecological belt, level of facilities, presence of external supervision, 24-hour duty schedule, quality assurance activities, number of total health workers and delivery beds were significantly associated with BEmONC service readiness score. Similarly, facilities that regularly reviewed maternal and newborn deaths had better readiness scores than the facilities not reviewing maternal and newborn deaths regularly.

In multiple linear regression, with reference to federal/provincial level hospitals, local HFs (β = -12.64, 95% CI: -18.31, -6.96, p-value: <0.001) and private hospitals have lower readiness score (β = -18.08, 95% CI: -24.08, -12.081, p-value: <0.001). Compared to Koshi, BEmONC readiness score is 3.38 points higher in Bagmati Province (95% CI: 0.36, 6.39, p-value: 0.029). Similarly, HFs located in rural settings had better service readiness score (β = 2.60, 95% CI: 0.62, 4.58, p-value: 0.010) than urban HFs. Facilities in the mountain region (β = 4.18, 95% CI: 1.65, 6.71, p-value: 0.001) and Terai belt (β = 3.22; 95%CI: 0.28, 6.15; p-value: 0.032)) had better readiness score than facilities in Hilly region. Some of the service management related variables were also associated with better readiness score, for example, facilities having external supervision (β = 2.99, 95% CI: 1.08, 4.89, p-value: 0.002), and 24 hours' duty schedule (β = 5.34, 95% CI: 2.84, 7.83, p-value: <0.001) and quality assurance activities (β = 3.59, 95%CI: 1.6, 5.54). With one unit increase in number of delivery bed, the readiness score increased by approximately one point (β = 0.996, 95% CI: 0.12, 1.88, p-value: 0.027).

## Discussion

The findings derived from NHFS 2021 provide significant and alarming insights into the state of BEmONC service readiness in Nepal. Findings reveal that the proportion of HFs with availability of all BEmONC signal functions is low particularly in local HFs with only <1% facilities having all BEmONC signal functions. This could be because only basic hospitals and selected other facilities at local level are envisioned to provide BEmONC level service in Nepalese

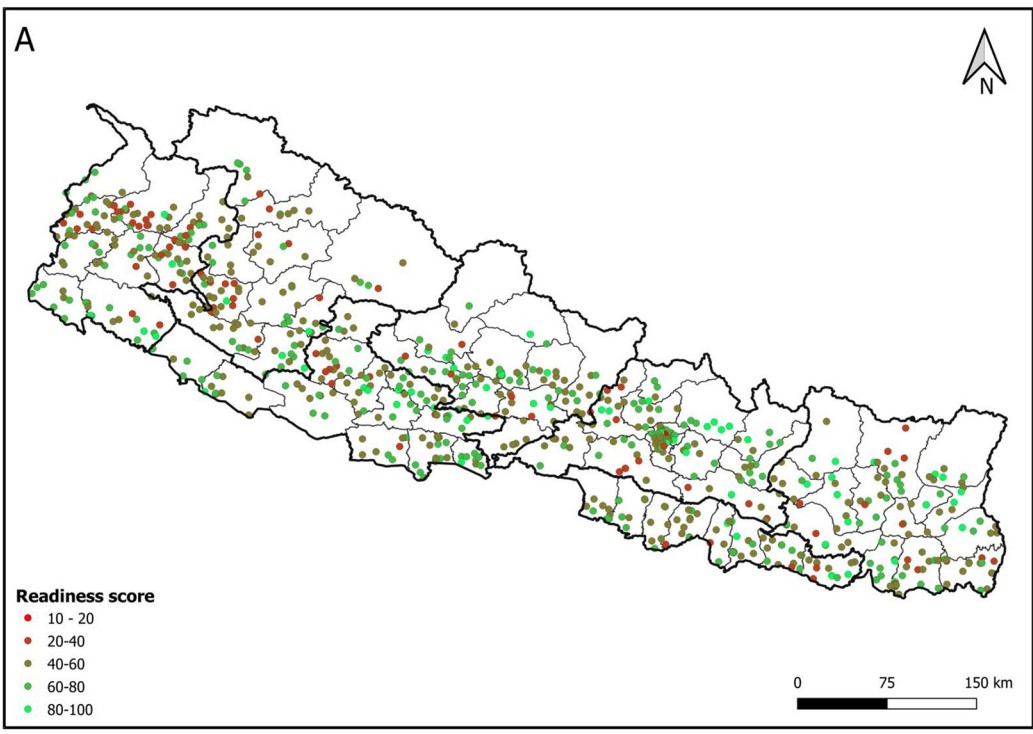

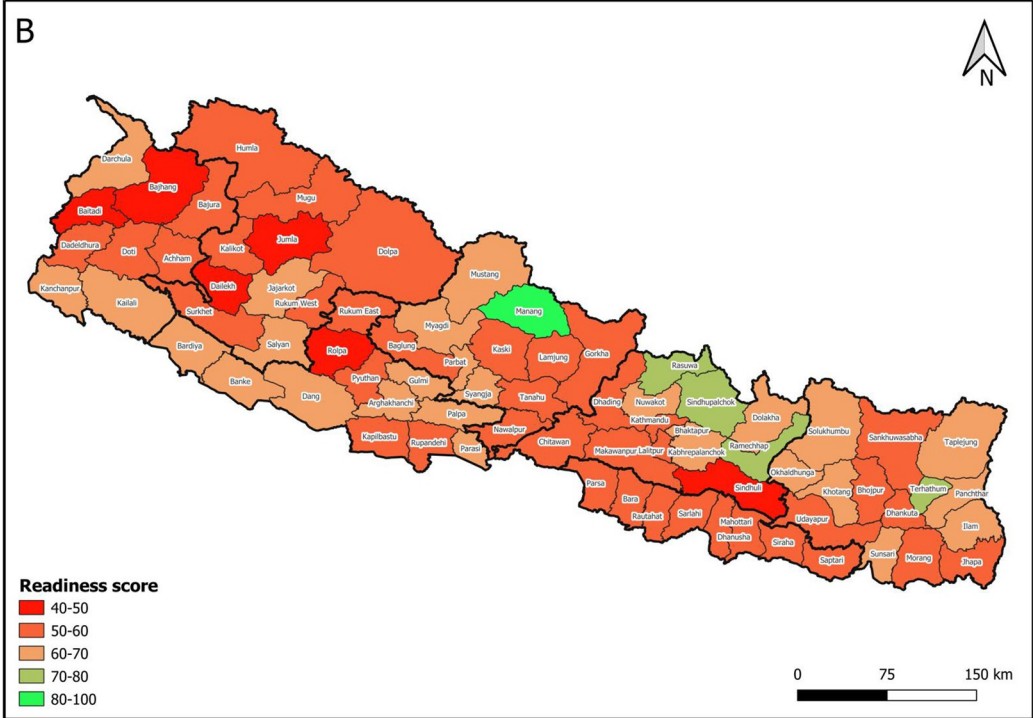

**Fig 3. Facility wise and district wise BEmONC readiness map was plotted in QGIS 3.22.7-Białowieża from freely available GPS dataset of the HFs from www.dhsprogram.com.** We used shape file with EPSG:4326 projection system publicly available at the official website of survey department of Ministry of Land Management, Cooperatives and Poverty Alleviation (Government of Nepal) (https://nationalgeoportal.gov.np/).

**Table 3. Comparison of readiness score for BEmONC services.**

| Domain | Overall n = 804 mean±sd; median (Q1, Q3) | Type of facilities | | |
|---|---|---|---|---|
| | | Federal/ provincial hospitals, n = 25 mean±sd; median (Q1, Q3) | Local HFs, n = 718 mean±sd; median (Q1, Q3) | Private hospitals, n = 61 mean±sd; median (Q1, Q3) |
| Guidelines and staff training | 21.2±28.2; 0.0 (0.0, 50.0) | 40.5±34.5; 50.0 (0.0, 50.0) | 21.7±28.1; 0.0 (0.0, 50.0) | 7.2±18.9; 0.0 (0.0, 0.0) |
| Essential equipment and supplies | 78.5±12.3; 78.6 (71.4, 85.7) | 94.2±7.7; 92.9 (92.9, 100.0) | 77.4±11.1; 78.6 (71.4, 85.7) | 84.2±19.4; 92.9(78.6, 100.0) |
| Essential medicines | 64.5±16.8; 63.6 (54.6, 72.7) | 84.0±14.0; 90.9 (72.7, 90.9) | 63.5±15.6; 63.6(54.5, 72.7) | 67.8±24.1; 72.7 (54.6, 81.8) |
| Overall readiness score | 54.7±13.4; 53.3 (45.0, 63.6) | 72.9±13.6; 74.6 (63.6, 80.3) | 54.2±12.8; 52.7 (45.0, 63.4) | 53.1±15.1; 55.2 (45.0, 61.3) |

sd: standard deviation; Q1: 1st quartile, Q3: 3rd quartile

context. However, to ensure the comparability of findings with studies in other countries and previous studies in Nepal, we have considered facility offering delivery service as denominator for the analysis. To offer additional insights on availability of BEmONC signal functions, we have disaggregated the data by type of facility.

Compared to results from NHFS 2015 [26], the availability of all other signal functions except parenteral oxytocin has decreased at national level which also resembles the findings presented in the annual report of DoHS [15]. One of the multicentric studies in 2022 has also revealed a considerable gap in neonatal care services which aligns with our study [27]. The met need of EOC has been continuously declining in the last four fiscal years. The met need of EOC was 12.6% in 2017/18, 11.6% in 2018/19, 11.1% in 2019/20, and 8.2% in 2020/21 [15]. When compared to other neighboring countries like Pakistan and Bangladesh, Nepal seems to have relatively low availability of BEmONC signal functions [28, 29]. For example, in Sindh Province (Pakistan), the proportion of facilities having parenteral antibiotics was 92%, oxytocin was 90%, service for manual removal of placenta was 92%, service for normal birth facilities was 82% and neonatal resuscitation service was 80% [28]. Similarly, 98% of HFs in Bangladesh had parenteral antibiotics, 90% had oxytocin, 86% had manual placenta removal service, and 92% had health workers who could remove residual products of conception [29]. The overall readiness score in case of Nepal is higher than that of some other countries like Tanzania and Kenya [30, 31]. Availability of BEmONC functions largely depended on the nature of health facility considered in the study, number of delivery load and difference in such factors across setting may be partly responsible for variations in availability of signal function in different countries. To illustrate, HFs with a high case load are more likely to encounter complicated cases on a regular basis, allowing them to be better prepared for such scenarios compared to HFs with a minimal number of delivery cases.

The aggregate weighted BEmONC service readiness score for federal/provincial hospitals, local HFs and private hospitals were 72.89, 54.21 and 53.09 respectively in 2021. In 2015, service readiness scores for federal/provincial hospitals, local HFs and private hospitals were 70.86, 51.59 and 49.62 respectively (based on further analysis from NHFS 2015, findings now reported in this manuscript), which indicate marginal decline between 2015 to 2021[19]. A study by Banstola et al. in conducted in 16 HFs of Taplejung district of eastern Nepal in 2018 showed the overall BEmONC readiness score to be 77% with the highest score for equipment (87%) and lowest for staff training and guidelines domains (51%) [32]. The readiness score is higher than the proportion reported in our study. When NHFS 2021 survey was undertaken, Nepal was battling COVID-19 pandemic and placing high priority on COVID-19 response and imposing restrictive measures including transportation. The survey was halted between May to July 2021 due to COVID-19-imposed lockdowns in Nepal. Some degree of disruption in supply chain, difficulties in mobility, and restriction in transportation measures with

**Table 4. Factors associated with BEmONC service readiness of HFs offering delivery and newborn care services.**

| Characteristic | Unadjusted | | Adjusted | |
|---|---|---|---|---|
| | β (95% CI) | p-value | β (95% CI) | p-value |
| **Province** | | | | |
| Koshi | Ref | | Ref | |
| Madhesh | -1.85 (-5.93, 2.24) | 0.400 | -3.05 (-7.43, 1.32) | 0.171 |
| Bagmati | 2.29 (-0.85, 5.44) | 0.200 | 3.38 (0.36, 6.39) | 0.029 |
| Gandaki | 1.05 (-2.54, 4.64) | 0.600 | 3.05 (-0.41, 6.51) | 0.085 |
| Lumbini | 2.21 (-1.02, 5.43) | 0.200 | 1.87 (-1.24, 4.98) | 0.239 |
| Karnali | -2.54 (-6.04, 0.96) | 0.200 | -1.73 (-5.09, 1.63) | 0.313 |
| Sudurpaschim | -1.08 (-4.35, 2.18) | 0.500 | -0.36 (-3.46, 2.73) | 0.819 |
| **Location** | | | | |
| Urban | Ref | | Ref | |
| Rural | 0.601 (-1.30, 2.50) | 0.500 | 2.60 (0.62, 4.58) | 0.010 |
| **Ecological region** | | | | |
| Hill | Ref | | Ref | |
| Mountain | 3.00 (0.43, 5.56) | 0.022 | 4.18 (1.65, 6.71) | 0.001 |
| Terai | 2.18 (-0.16, 4.52) | 0.068 | 3.22 (0.28, 6.15) | 0.032 |
| **Level of facility** | | | | |
| Federal/Provincial hospital | Ref | | Ref | |
| Local HFs | -18.67 (-23.92, -13.43) | <0.001 | -12.64 (-18.31, -6.96) | <0.001 |
| Private Hospital | -19.8019 (-25.9230, -13.6808) | <0.001 | -18.08 (-24.08, -12.08) | <0.001 |
| **External supervision** | | | | |
| No | Ref | | Ref | |
| Yes | 3.57 (1.57, 5.57) | <0.001 | 2.99 (1.08, 4.89) | 0.002 |
| **Duty scheduled in 24 hours** | | | | |
| No | Ref | | Ref | |
| Yes | 6.71 (4.49, 8.94) | <0.001 | 5.34 (2.84, 7.83) | <0.001 |
| **Quality assurance activities** | | | | |
| Not Performed | Ref | | Ref | |
| Performed | 4.7247 (2.72, 6.73) | <0.001 | 3.59 (1.64, 5.54) | <0.001 |
| **Review maternal and newborn deaths** | | | | |
| No | Ref | | Ref | |
| Yes | 2.78 (0.79, 4.77) | 0.006 | 0.36 (-1.60, 2.32) | 0.719 |
| **System for review of client's opinion** | | | | |
| No | Ref | | Ref | |
| Yes | 3.66 (-0.51, 7.83) | 0.085 | 0.10 (-4.0, 4.20) | 0.962 |
| **Total health workers** | 0.017 (0.01, 0.03) | <0.001 | 0.01 (-0.005, 0.018) | 0.263 |
| **Number of delivery beds** | 1.93 (1.07, 2.79) | <0.001 | 0.996 (0.12, 1.88) | 0.027 |

CI: Confidence Interval; Ref: Reference group; β: beta coefficients

inadequate preparation to ensure the supply of health commodities could be responsible for lower availability of medicines/ equipment and thus leading to lower readiness score. One of the previous studies has also indicated that maternal and newborn services were significantly impacted by COVID-19 pandemic and pointed out the need to improve access to high quality intrapartum care to prevent excess death [33].

In contrast to a previous study that analyzes findings from NHFS 2015 [19], our study found that the readiness of BEmONC services was better in rural areas, with a statistically

significant difference of 2.76 points (p-value = 0.008). This improvement in service readiness in rural areas could be because of the changes in service delivery that have taken place in Nepal since 2015. It is important to note that urban areas offer multiple alternative options and choices, such as higher-level medical facilities, private hospitals, and medical colleges. Considering the limited choices available in rural settings, provincial and local governments may have prioritized training of healthcare workers in rural settings, which seems to be driving the overall readiness score in rural settings.

Among individual domains of readiness score, guidelines and staff training domain have the lowest score in each type of HFs and thus have the highest scope for improvement. Also, the study reveals that only one third of HFs have at least one health service provider trained on delivery and newborn services like SBA, advanced SBA, maternal and newborn health update, routine care during labor and normal vaginal delivery, and active management of third stage labor (AMTSL) in last 24 months preceding the survey. This could be because only a few training sessions were organized after the emergence of COVID-19 pandemic due to restrictive measures limiting gathering of health workers in one place. Further, redefinition of roles and responsibilities in federal structure and limited understanding among the concerned authorities could be some other reasons. The capacity development activities can be further accelerated in collaboration between seven Provincial Health Training Centers (PHTC) and 62 training sites distributed across seven provinces [15]. On-site coaching and clinical skill enhancement have been found as effective strategies to improve knowledge, skills and practice of service providers [15]. PHTC can also collaborate with provincial hospitals and academies to further accelerate the on-site coaching and mentoring related to BEmONC level care. As per the new federal structure, local level governments are responsible for delivery of basic health services and facilities under local government are front line facilities providing service to people. Regular monitoring of service delivery sites, identifying challenges in service availability and addressing them in coordination between local, provincial and federal government could be an effective strategy. Cascade-mentoring approach [34], which involves training a cadre of local mentors who can then receive further guidance and support from mentors at the provincial level, can help accelerate the mentorship process in Nepal [35]. One of the previous studies indicates that almost one third of the decline in maternal and neonatal mortality (28% decline in each case) could be realized through improvement in quality of care in low- and middle-income countries [36]. Promoting quality improvement initiatives through self-assessment was found to be effective in a pilot study conducted in Taplejung and Hetauda and its expansion in BEmONC level facilities across the country could be effective [15].

Findings of our study present a comprehensive understanding of where Nepal stands in terms of availability of BEmONC signal functions and readiness for BEmONC services. As the decision-making authority has been delegated to 7 provincial and 753 provincial governments, there could be opportunities for improving readiness, BEmONC signal functions availability and quality of service in all birthing centers. Most of the responsibilities previously discharged through District Health Offices are now being delivered through municipality offices. Health workers, particularly paramedics leading health sections in local governments, are primarily trained to deliver healthcare services and but are not capacitated enough for planning, governance and financial management. Health coordinators/chiefs of the local governments need to be trained in healthcare delivery planning, logistic management, monitoring, evaluation, and general management [37] to improve the overall service availability and readiness. By improving local governments' ability to collect and use evidence, it may be feasible to use their decision-making authority for more context-specific planning, resulting in an increase in the quality and breadth of services, including BEmONC services [37].

Further studies that involve in-depth assessment of birthing centers in each province with cost estimation for upgrading HFs as per the national standards could be useful. Apart from improvement in service readiness and availability, efforts should also be directed to raise awareness at the community level to encourage utilization of available health services.

This study involves further analysis of nationally representative NHFS 2021 that considers the federal structure during sampling procedure, that ensures generalizability across all seven provinces in the country. Secondly, the study utilized the standardized SARA manual from the WHO to assess the readiness and availability of HFs for BEmONC services. This standardized tool enables comparability of the study's findings in other countries across the world. Third, the measurement of availability of items were based on reporting by health facility members as well as observation by enumerator which prevents the over reporting by the HFs.

There are some limitations to this study. First, we utilized data from a survey that was designed to assess the service availability and readiness for multiple other conditions like family planning, tuberculosis, animal bite, non-communicable diseases and was not primarily dedicated to BEmONC services. As the survey was not primarily dedicated to the study of BEmONC services, multiple other variables like delivery load and number of cases with complication have not been included in this study. Second, this study was undertaken during the period of COVID-19 pandemic, so some components of service availability and readiness might have been under-estimated or over-estimated because of resource diversion in COVID-19 response and restrictive measures that disrupted supply chain to some level. Third, this study doesn't account for the quality of the services, which could be important considering health outcomes. Finally, the cross-sectional nature of the study may mask situations where equipment, medicine, and commodities were generally available most of the time in year but temporarily unavailable during the study and vice-versa.

## Conclusions

The availability of all seven BEmONC signal functions and readiness of HFs for BEmONC services were relatively low, particularly in local HFs and private hospitals. HFs having external supervision, quality assurance activities and presence of duty scheduled for 24 hours have better readiness score for BEmONC services. Findings suggest that, together with expansion, Nepal needs to pay attention to quality of services ensuring availability of trained health workers, medicines and equipment for maternal and newborn care.

## Supporting information

**S1 Table. Detail of tracer items and readiness score calculation process.**
(XLSX)

**S2 Table. Definition of independent variables.**
(DOCX)

**S3 Table. Availability of seven signal functions.**
(DOCX)

## Acknowledgments

We would like to acknowledge United States Agency for International Development's Demographic Health Survey program for providing us the datasets and thank all who directly and indirectly supported us in this study.

## Author Contributions

**Conceptualization:** Achyut Raj Pandey, Sushil Chandra Baral.

**Data curation:** Achyut Raj Pandey.

**Formal analysis:** Bikram Adhikari.

**Methodology:** Achyut Raj Pandey, Bikram Adhikari.

**Supervision:** Sushil Chandra Baral.

**Validation:** Achyut Raj Pandey, Bipul Lamichhane.

**Visualization:** Bikram Adhikari.

**Writing – original draft:** Achyut Raj Pandey, Deepak Joshi.

**Writing – review & editing:** Bikram Adhikari, Bipul Lamichhane, Deepak Joshi, Shophika Regmi, Bibek Kumar Lal, Sagar Dahal, Sushil Chandra Baral.

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
