## [Decision Letter · Decision Letter 0]

13 Mar 2023

PONE-D-23-04260Service Availability and Readiness for Basic Emergency Obstetric and Newborn Care: Analysis from Nepal Health Facility Survey 2021PLOS ONE

Dear Dr. Pandey,

Thank you for submitting your manuscript to PLOS ONE.There are number of comments that the reviewers' have raised, please ensure all the comments are responded in particular reviewer 3's comment.After careful consideration, we invite you to submit a revised version of the manuscript that addresses the points raised during the review process.

We look forward to receiving your revised manuscript.

Kind regards,

Ashish KC, PhD, Associate Professor, Department of Women's and Children's Health, Uppsala University, Sweden

Academic Editor

PLOS ONE

3. We note that Figure 4 in your submission contain [map/satellite] images which may be copyrighted. All PLOS content is published under the Creative Commons Attribution License (CC BY 4.0), which means that the manuscript, images, and Supporting Information files will be freely available online, and any third party is permitted to access, download, copy, distribute, and use these materials in any way, even commercially, with proper attribution. For these reasons, we cannot publish previously copyrighted maps or satellite images created using proprietary data, such as Google software (Google Maps, Street View, and Earth). For more information, see our copyright guidelines: http://journals.plos.org/plosone/s/licenses-and-copyright.

a. You may seek permission from the original copyright holder of Figure 4 to publish the content specifically under the CC BY 4.0 license. 

Reviewers' comments:

Reviewer's Responses to Questions

**Comments to the Author**

1. Is the manuscript technically sound, and do the data support the conclusions?

Reviewer #1: Yes

Reviewer #2: Yes

Reviewer #3: Partly

2. Has the statistical analysis been performed appropriately and rigorously? 

Reviewer #1: Yes

Reviewer #2: Yes

Reviewer #3: I Don't Know

3. Have the authors made all data underlying the findings in their manuscript fully available?

Reviewer #1: No

Reviewer #2: Yes

Reviewer #3: Yes

4. Is the manuscript presented in an intelligible fashion and written in standard English?

Reviewer #1: Yes

Reviewer #2: Yes

Reviewer #3: Yes

5. Review Comments to the Author

Reviewer #1: The full data set was not present, but I presume it can be added as part of the revision. While the manuscript was generally well constructed, addressing the major points of the review may require reanalysis of the data set.

Reviewer #2: 1.In Abstract, Methods section-sentence no. 3, It would be better to use inferential statistics to cover mean, standard deviation, median and interquartile range…

2.In Abstract Section, Results-1st sentence, mention is Madesh and Gandaki are Facilities or Province or what is appropriate

3.The last sentence of the result section in abstract looks more like a conclusion, could the figures from your result table be added here instead

4.In introduction, 3rd paragraph, it is advisable to mention a full form for SMNH and EOC.

5.Likewise for PHCC, CHU, HP etc towards the last paragraph, please do mention their respective fullforms.

6.Introduction, last para, 3 rd line, “compared to no skill’ can be omitted

7.For Methodology section, is it possible to build a STROBE flow diagram to make things more clear at a glance.

Reviewer #3: Thank you for the opportunity to review this paper on Basic Emergency Obstetric and Newborn Care in Nepal using data from the National Health Facility Survey 2021. It is an important topic and reveals low levels of health facility readiness in Nepal to provide BEmONC services by using a “readiness score” across three domains (guidelines/staff, equipment/supplies, and medicines/commodities). The paper in the current form has some notable gaps, which are flagged below. With no page numbers or line numbers, I could not give specific but have provided comments by section of the paper for the authors to consider. In the future, page and line numbers should be added to assist the reviewer.

Abstract: the results are hard to interpret without understanding what proportion of facilities would be expected to have all signal functions (is this out of 100) and details about readiness score.

Introduction

- There is not consistency on focus across maternal and newborn continuum. You start by focusing on the maternal mortality SDG, with no mention of the newborn mortality target or stillbirth target (not an SDG but is an Every Newborn target), and then link EMoNC to newborn mortality in the final paragraph without referencing evidence about impact of EMoNC on maternal or other birth outcomes.

- You use the term EOC without spelling this out (assuming is Emergency Obstetric Care). You may want better define these terms and what is difference between EOC, EmONC, BEmONC, CEmONC for the broader audience of PLOS ONE.

Methods:

- It would help the reader to explain more about the health system and facility levels under the sample section. You list the different groups of facilities but it is not clear what services, including BMoNC or not, would be expected at each. Also in the final sentence of this section indicates that you only considered facilities that offered normal vaginal deliveries. Does this include facilitates that office BEmONC?

- More details on data collection are needed in this paper to allow the reader to understand who collected the data and how it was collected (without having to go to another report or paper). For example, under variables and measurements, you indicate “The health facility was defined to have availability of functional signal function of delivery and newborn care if the service was available and provided within past 3 months preceding the survey.” However, you have not explained how it is determined – self-reporting through surveys, observation from facility visits, etc.. This is the same for the other variables and measurement sections. For example, later on page 12, you have indicated “Facilitates are said to have” – said by whom? The data collection overall needs to be clearer and also added to the strengths and limitation in the discussion section.

- Please include clearly how you calculated the readiness scores. A reader should not have to go to the WHO manual to understand scoring. You describe the scoring in the results section noting that these domains each are on a scale to 33.3333 but this belongs in the methods section. You should also explain why these domains are considered of equal weight for the full score.

Results

- what is the rationale for presenting the unweighted and weighted data (Table 1&2) and what do these different data tell us? Similarly unadjusted and adjusted (Table 3). This should be well explained in methods and interpreted in the narrative of the results text.

- some of the results discussed in the narrative text are not significant and could be misleading eg “facilities having system for review of maternal and newborn deaths had better readiness score than those not having such review.”

- the authors should make use of supplementary files to provide further details for methods and results.

Discussion

The study did not assess trends and yet this is a major focus of the discussion.

Some important literature on quality improvement for MNH care broadly and specifically in Nepal is absent in the discussion and interpretation of the results thus it is difficult to see how this work contributes to the literature of what is not known already

OTHER

General grammar and typos: there are quite a few sentences that are not easy to read as well as some minor typos (eg forgot a period at the end of the sentence). You should consider a copy edit. Additionally, not all of the acronyms are spelled out for the reader eg MOHP, ICF, PHCC, HPs, CHUs, HTCs, UHCs

References are not complete or in line with PLOS ONE format. Reports should include a website and access date, when available.

Ethics statement: more details are needed - even for secondary analysis you need to add what ethical clearance was received for the primary analysis.

Data availability: please provide the exact webpage and not just the DHS general webpage to direct readers as to the source.

6. PLOS authors have the option to publish the peer review history of their article (what does this mean?). If published, this will include your full peer review and any attached files.

Reviewer #1: **Yes: **Robert B Clark

Reviewer #2: No

Reviewer #3: No

---

## [Author Response · Author response to Decision Letter 0]

27 Apr 2023

Response to reviewers comment

Response: The manuscript ensures the PLOS One style requirements.

Response: Thank you for the feedback. We have addressed this comment in the manuscript. This study is further analysis of NHFS 2021, and no separate ethical clearance is required for further analysis in Nepalese context. In the original survey, ethical clearance was obtained from Ethical Review Board of Nepal Health Research Council. Written informed consent was obtained from all participants after explaining about risk benefits and voluntariness of participation. [line 227-232]

3. We note that Figure 4 in your submission contain [map/satellite] images which may be copyrighted. All PLOS content is published under the Creative Commons Attribution License (CC BY 4.0), which means that the manuscript, images, and Supporting Information files will be freely available online, and any third party is permitted to access, download, copy, distribute, and use these materials in any way, even commercially, with proper attribution. For these reasons, we cannot publish previously copyrighted maps or satellite images created using proprietary data, such as Google software (Google Maps, Street View, and Earth). For more information, see our copyright guidelines: http://journals.plos.org/plosone/s/licenses-and-copyright.

Ans: Thank you for the feedback. The maps presented in the manuscript are copyright free. Map was plotted in QGIS 3.22.7-Białowieża from freely available GPS dataset of the health facilities from www.dhsprogram.com. We used a shape file with the EPSG:4326 projection system publicly available at the official website of Survey department of Ministry of Land Management, Cooperatives and Poverty Alleviation (Government of Nepal) (https://nationalgeoportal.gov.np/). The software used and source of shape files are cited in the current version of map. [Line 222-226]

 

Response to comments from reviewer-1, Robert B. Clark, MD

Overall: Excellent analysis which provides valuable guidance to policymakers and administrators regarding gaps in maternal/newborn care. Robust review with facility capacity in Nepal, emphasizing signal functions and preparedness. Appropriate recommendations regarding gaps and lack of progress in some areas. 

Major Points to Consider:

1. A major point is the terminology employed throughout the manuscript, which is confusing at times. The 7 signal functions are evaluated. The preparedness of facilities are evaluated. “Availability” and “readiness” are not consistently used with these two terms and are overlapping terms. Further, the components overlap, as with uterotonic. Clear delineations between these two groups of indicators would be helpful. Consider using “performed” or “presence” or similar to describe 7 signal functions (or always state “Availability of BEmONC signal functions” as the default). Consider using “preparedness” or “readiness (exclusively) to describe the three domains. 

Response: We have corrected and have maintained uniformity in terminologies used as per the suggestion provided. We agree that there could be alternative ways the variables could be identified and used in the study. However, we have considered the ‘Service Availability and Readiness Assessment (SARA) Manual by World Health Organization to facilitate comparison of findings with previous findings from Nepal and that from other countries. 

2. With the signal functions, a major point with the results and the discussion is the lack of analysis according to delivery burden of the facility. USAID concludes “only facilities with a case load of at least 250 births in three months have a reasonable chance to encounter the complications to provide EmONC signal functions.” It is therefore unlikely that the smaller facilities would have performed all signal functions in the last three months. The percentage of facilities with > 250 births per month will all (or partial) signal functions would be more meaningful from standpoint of addressing actionable gaps in capacity. 

Response: Thank you for the feedback. We agree that the need for BEmONC signal functions depends on the number of deliveries performed per month or three months. The study involves further analysis of Nepal Health Facility Survey, that does not have variable on number of deliveries performed per month. We made calculations based on SARA tool of WHO. However, to address the feedback, we have presented disaggregated analysis by facility type for readiness score and service availability (Table number 1, 2,3 and Figure: 2). The federal /provincial hospitals (including district hospitals) are the facilities that are exclusively supposed to have BEmONC signal functions as per the policy documents in Nepal. However, to comply with the calculations performed in Health Facility Survey Report, SARA tool and other previous publications, we also have data analysis performed at aggregate level. 

3. Similarly, for all the facilities with less than 250 births per month, it would be expected that they would not have performed all signal functions, but knowing the ones that have been most commonly used can still be instructive. 

Response: Thank you for the feedback. We agree that the need for BEmONC signal functions depends on the number of deliveries performed per month or three months. The study involves further analysis of Nepal Health Facility Survey, that does not have variable on number of deliveries performed per month. We made calculations based on SARA tool of WHO. However, to address the feedback, we have presented disaggregated analysis by facility type for readiness score and service availability (Table number 1, 2,3 and Figure: 2). The federal /provincial hospitals (including district hospitals) are the facilities that are exclusively supposed to have BEmONC signal functions as per the policy documents in Nepal. However, to comply with the calculations performed in Health Facility Survey Report, SARA tool and other previous publications, we also have data analysis performed at aggregate level. 

4. The low score of the guideline and staff domain is discussed in detail. Value could be added with a list of recommendations for the most time-efficient options for addressing this specific gap. What training or courses does NHTC offer that is a refresher for AMSTL? Or AVD or resuscitation, etc.? How can this specific gap be expeditiously addressed? 

Response: We have added some contents in the discussion part of manuscript. NHTC covers issues or skills like estimation of blood loss and prevention of PPH in AMTSL training. Similarly, NHTC also conducts other training in maternal and newborn health according to need and in consultation with Family Welfare Division. There are seven provincial health training centres and 62 training sites scattered in seven provinces in Nepal. NHTC could further accelerate the refresher training through coordination and collaboration with the PHTCs and training sites. Further, regular training need assessment with active involvement of Provincial Health Directorate and addressing those gaps in collaboration between NHTC, PHTC and training sites.

5. Similarly, the authors could select 2-3 other specific major gaps and give specific guidance on solving the problem, instead of simply stating the problem exists. 

Response: Thank you for the feedback. We have addressed feedback in throughout the discussion section.

6. The conclusion lacks specificity. It summarizes the study adequately, but this summary does not significantly add to what is already known from previous studies. 

Response: Thank you for the feedback, We have re-written the conclusion sections (Line: 437-442)

Minor Points to Consider:

Abstract

• Minor grammar corrections are needed throughout the manuscript.

Response: We have checked grammar throughout the manuscript and have made corrections accordingly

• Readiness score scale of 100 should be mentioned. 

Response: The comment has been addressed [Line:21]

• Conclusions are unclear and perhaps miss the most important points.

Response: We have revised the conclusion section [Line: 49-44]

Background

• 1st paragraph – inconsistencies regarding MMR decreases

Response: We have corrected inconsistencies in the manuscript. 

• NME should be addressed more fully, such as in the first paragraph.

Response: Thank you for the feedback. The contents on NMR has been added in the first paragraph as per suggestion

• Case rate fatality seems out of place as it is not addressed elsewhere. 

Response: Thank you for the feedback. It has been removed [49-60]

Methods

• Six groups – unclear if the six groups are comprised of each of the types or 6 mixed groups. 

Response: The sentence is rephrased for clarity. The facilities were classified into six categories: Hospitals, Primary Health Care Centers (PHCC), Health Posts (HP), Community Health Units (CHUs), stand-alone HIC Testing and Counselling (HTCs), and Urban Health Centers (UHCs). [Line 135-139] 

• Data collection – a comment regarding the rationale for the non-use of the other data sources would be helpful. 

Response: Thank you for the feedback. To facilitate comparison with previous publications from Nepal and publications from other countries, we have adopted ‘Service Availability and Readiness Assessment (SARA) guideline developed by World Health Organization. SARA tool is based on NHFS and we opted for analysis of NHFS in our study. Further, NHFS being the only health facility survey in Nepal, we did not have option to include any other data sources. [Line: 102-103]

• Service readiness – readers can also view table 4 for a list. The supplemental information could include further information regarding items, which would be helpful, i.e., what comprises Emergency Transport? Delivery package? Does resuscitation mask also include bag?

Response: The definition of the different tracer items is made available in supplementary file 1. We missed resuscitation bag while writing but Resuscitation bag and mask are included in that item while analysis which is now edited and corrected in the manuscript [Table 2]

• Monitoring or supervision – does this refer to clinical or administrative supervision?

Response: Monitoring and supervision refers to both clinical and administrative supervision.

• Statistical analysis – Is there a list of the continuous variables? Were the same analyses performed for the availability of the signal functions?

Response: Readiness score, number of beds and number of health care workers are the continuous variables. 

Results

• The overlap of oxytocin and injectable uterotonic is confusing, since in essentially all cases these are the same. Consider dropping uterotonic from analysis. 

Response: We adopted the variables names in accordance with ‘WHO Service availability and readiness assessment manual (available at: (https://www.who.int/publications/i/item/WHO-HIS-HSI-2014.5-Rev.1) and previous publication (https://pubmed.ncbi.nlm.nih.gov/34288943/). 

• The context of population would be very helpful, such as % of population who lives in Hilly or Mountain regions, % of pop. in each province. 

Response: This part is now added in the study setting part of the methods. [Line108-118]

• SNCU is not mentioned earlier as a variable.

Response: This part is removed to reduce confusion.

• Due to the delivery load concern, Figure 2 and the paragraph preceding it that analyzed the signal functions were not that useful. Further, the figure following the descriptive paragraph seems a bit redundant. 

Response: Thank you for the feedback. Figure has been removed as it does not yield any additional information. The redundant paragraph has also been removed.

• The value of number of beds and health workforce is questionable, since it has no context to the size of the facilities. Similarly, the presence of SNIC/NICU would not be expected in lower-level facilities, making this observation difficult to interpret. 

Response: Thank you for the feedback. NHFS does not have variable on number of deliveries. However, we have disaggregated facilities by level (local, provincial/federal and private) which we assume clarifies the issue. Provincial and Federal level facilities are expected to have SNCU /NICU level service as per government policy.[Table 1]

• Figure 4 has excellent information. Unfortunately, the wise economy of combining the data into a single figure makes it difficult to clearly see the colors of the facilities wise scores.

Response: We agree that the map has been complex to understand due to loading of multiple information in the same map. We have not split the map into two to make it clearer. (Figure 3)

• Since the G-A estimation plot analysis shows little difference in the scores, Figure 5 may not be that useful. A description of the analysis may suffice. 

Response: The G-A estimation plot (Figure-5) has been removed

Discussion

• The decline in signal functions is important, as are possible reasons for the decline. However, this decline is overshadowed by the lack of delivery load consideration. Is the decline still evident if only facilities with 250/month are considered?

Response: In order prevent overshadowing due to lack of delivery load consideration, we calculated readiness score and determine BEmONC signal function availability by type of facility (federal/provincial hospital, local health facilities and private hospitals)

• Comparing the signal function with other countries also requires consideration of delivery load. Bangladesh is densely populated and may not have as many small delivery units as Nepal. 

Response: Thank you for the feedback. We have addressed the concern

• The federal and covid challenges are excellent points. 

Response: Thank you

Conclusion

• The message of little progress in signal function performance since 2015 is important – if there really is not progress in the facilities with large delivery loads. This deserves more attention after reanalysis. The same is true for the domains of preparedness. The conclusion might be an excellent place to restate the most severe gaps identified, with reference to concrete potential solutions described in the discussion. 

• Response: Thank you for the feedback. We have revised the conclusion section [Line 433-438]

 

Response to comments from reviewer-2

Please find the following to be considered for revision.

1.In Abstract, Methods section-sentence no. 3, It would be better to use inferential statistics to cover mean, standard deviation, median and interquartile range

Response: Thank you for your suggestion, we presented mean, SD, median and interquartile range 

2.In Abstract Section, Results-1st sentence, mention is Madesh and Gandaki are Facilities or Province or what is appropriate

Response: Thank you for the feedback, we have removed the sentences and replaced with other to address other comments from other reviewers.

3.The last sentence of the result section in abstract looks more like a conclusion, could the figures from your result table be added here instead

Response: Thank you reviewers for your comment, we have rewrite the last paragraph of result section of abstract [line 30-37]

4.In introduction, 3rd paragraph, it is advisable to mention a full form for SMNH and EOC.

Response: We have added full form of the SMNH and EOC in the manuscript

5.Likewise for PHCC, CHU, HP etc towards the last paragraph, please do mention their respective full forms.

Response: The full forms of the different acronyms are added throughout the manuscript wherever necessary.

6.Introduction, last para, 3 rd line, “compared to no skill’ can be omitted

Response: Thank you, we have omitted the phrase in the current version of manuscript. [Line 91-92]

7.For the Methodology section, is it possible to build a STROBE flow diagram to make things clearer at a glance.

Response: we have added STROBE’s flow diagram explaining sampling strategy and sample selection. [Figure 1] 

Reviewer-3

Thank you for the opportunity to review this paper on Basic Emergency Obstetric and Newborn Care in Nepal using data from the National Health Facility Survey 2021. It is an important topic and reveals low levels of health facility readiness in Nepal to provide BEmONC services by using a “readiness score” across three domains (guidelines/staff, equipment/supplies, and medicines/commodities). The paper in the current form has some notable gaps, which are flagged below. With no page numbers or line numbers, I could not give specific but have provided comments by section of the paper for the authors to consider. In the future, page and line numbers should be added to assist the reviewer.

Response: We have added line number in this current version of manuscript

1. Abstract: the results are hard to interpret without understanding what proportion of facilities would be expected to have all signal functions (is this out of 100) and details about readiness score.

Response: Methods part of abstract now contains how the score is calculated. [Line 19-23]

2. Introduction- There is not consistency on focus across maternal and newborn continuum. You start by focusing on the maternal mortality SDG, with no mention of the newborn mortality target or stillbirth target (not an SDG but is Every Newborn target), and then link EMoNC to newborn mortality in the final paragraph without referencing evidence about impact of EMoNC on maternal or other birth outcomes.- You use the term EOC without spelling this out (assuming is Emergency Obstetric Care). You may want better to define these terms and what is difference between EOC, EmONC, BEmONC, CEmONC for the broader audience of PLOS ONE.

Response: Thank you for the feedback. We have revised the manuscript as per the feedback.

3. Methods:

- It would help the reader to explain more about the health system and facility levels under the sample section. You list the different groups of facilities, but it is not clear what services, including BMoNC or not, would be expected at each. Also in the final sentence of this section indicates that you only considered facilities that offered normal vaginal deliveries. Does this include facilitates that office BEmONC?

Response: We have added about health system of Nepal before sampling section. We have included all facilities that offers delivery and newborn care services. The sentences are now rephrased for clarity. [Figure 1, line 119-126]

- More details on data collection are needed in this paper to allow the reader to understand who collected the data and how it was collected (without having to go to another report or paper). For example, under variables and measurements, you indicate “The health facility was defined to have availability of functional signal function of delivery and newborn care if the service was available and provided within past 3 months preceding the survey.” However, you have not explained how it is determined – self-reporting through surveys, observation from facility visits, etc. This is the same for the other variables and measurement sections. For example, later on page 12, you have indicated “Facilitates are said to have” – said by whom? The data collection overall needs to be clearer and added to the strengths and limitations in the discussion section.

Response: The details of data collection and operational definition are edited to make it more clearer. [Line 185-208, 145-153]

- Please include clearly how you calculated the readiness scores. A reader should not have to go to the WHO manual to understand scoring. You describe the scoring in the results section noting that these domains each are on a scale to 33.3333 but this belongs in the methods section. You should also explain why these domains are considered of equal weight for the full score.

Response: The method of calculation of readiness score is mentioned in the method section of the manuscript and the detailed process of calculation is present in the supplementary file 1.

4. Results- what is the rationale for presenting the unweighted and weighted data (Table 1&2) and what do these different data tell us? Similarly unadjusted and adjusted (Table 3). This should be well explained in methods and interpreted in the narrative of the results text.

Response: Unweighted analysis part is removed from throughout the manuscript.

- some of the results discussed in the narrative text are not significant and could be misleading eg “facilities having system for review of maternal and newborn deaths had better readiness score than those not having such review.”

Response: We have revised the results sections

- the authors should make use of supplementary files to provide further details for methods and results.

Response: We have added supplementary files providing details of methods and results [Supplementary file 1]

5. Discussion: The study did not assess trends and yet this is a major focus of the discussion.

Response: We have analyzed NHFS 2015 and compared readiness score and BEmONC services availability between 2015 and 2021. [Supplementary file 3 and 4, Line 312-318, 270-275, Supplementary file 3]

Some important literature on quality improvement for MNH care broadly and specifically in Nepal is absent in the discussion and interpretation of the results thus it is difficult to see how this work contributes to the literature of what is not known already

Response: Thank you for the feedback. We have revised the discussion section to address the concern

6. OTHER: General grammar and typos: there are quite a few sentences that are not easy to read as well as some minor typos (eg forgot a period at the end of the sentence). You should consider a copy edit. Additionally, not all of the acronyms are spelled out for the reader eg MOHP, ICF, PHCC, HPs, CHUs, HTCs, UHCs

Response: Full form for all acronyms were added throughout the manuscript.

References are not complete or in line with PLOS ONE format. Reports should include a website and access date, when available.

Response: Thank you for the feedback. We have revised the reference sections as per PLOS One format

Ethics statement: more details are needed - even for secondary analysis you need to add what ethical clearance was received for the primary analysis.

Response: The ethical statement was explained in detailed in the current version of manuscript [Line number:227-232]

Data availability: please provide the exact webpage and not just the DHS general webpage to direct readers as to the source.

Response: The exact website for downloading data is https://dhsprogram.com/data/available-datasets.cfm which is now updated in the manuscript. [Line: 445-447]

---

## [Decision Letter · Decision Letter 1]

13 Jun 2023

PONE-D-23-04260R1Service Availability and Readiness for Basic Emergency Obstetric and Newborn Care: Analysis from Nepal Health Facility Survey 2021PLOS ONE

Dear Mr. Pandey

Thank you for your revision and effort. You have made significant improvement in the revised version. Based on the third reviewer's comment, further revisions are required. I hope to get a revised version soon. This work of your will be a substantial contribution to global maternal and newborn health literature.

warm regards, Ashish Please submit your revised manuscript by Jul 28 2023 11:59PM. If you will need more time than this to complete your revisions, please reply to this message or contact the journal office at plosone@plos.org. Please include the following items when submitting your revised manuscript:A rebuttal letter that responds to each point raised by the academic editor and reviewer(s). You should upload this letter as a separate file labeled 'Response to Reviewers'.A marked-up copy of your manuscript that highlights changes made to the original version. You should upload this as a separate file labeled 'Revised Manuscript with Track Changes'.An unmarked version of your revised paper without tracked changes. You should upload this as a separate file labeled 'Manuscript'.If applicable, we recommend that you deposit your laboratory protocols in protocols.io to enhance the reproducibility of your results. Protocols.io assigns your protocol its own identifier (DOI) so that it can be cited independently in the future. For instructions see: https://journals.plos.org/plosone/s/submission-guidelines#loc-laboratory-protocols. Additionally, PLOS ONE offers an option for publishing peer-reviewed Lab Protocol articles, which describe protocols hosted on protocols.io. Read more information on sharing protocols at https://plos.org/protocols?utm_medium=editorial-email&utm_source=authorletters&utm_campaign=protocols.

We look forward to receiving your revised manuscript.

Kind regards,

Ashish KC

Academic Editor

PLOS ONE

Journal Requirements:

Reviewers' comments:

Reviewer's Responses to Questions

**Comments to the Author**

1. If the authors have adequately addressed your comments raised in a previous round of review and you feel that this manuscript is now acceptable for publication, you may indicate that here to bypass the “Comments to the Author” section, enter your conflict of interest statement in the “Confidential to Editor” section, and submit your "Accept" recommendation.

Reviewer #1: (No Response)

Reviewer #2: All comments have been addressed

Reviewer #3: All comments have been addressed

2. Is the manuscript technically sound, and do the data support the conclusions?

Reviewer #1: Yes

Reviewer #2: Yes

Reviewer #3: Partly

3. Has the statistical analysis been performed appropriately and rigorously? 

Reviewer #1: Yes

Reviewer #2: Yes

Reviewer #3: I Don't Know

4. Have the authors made all data underlying the findings in their manuscript fully available?

Reviewer #1: Yes

Reviewer #2: Yes

Reviewer #3: Yes

5. Is the manuscript presented in an intelligible fashion and written in standard English?

Reviewer #1: Yes

Reviewer #2: No

Reviewer #3: No

6. Review Comments to the Author

Reviewer #1: Overall: There is a marked improvement in the clarity of the report and the prior concerns are well-addressed. This study remains an excellent analysis that can provide needed guidance to decision makers in Nepal.

Several minor points can use additional consideration:

• Interpretation of the availability of signal functions is still problematic due to the mixture of low and high delivery loads at the Local HF. Further insight is limited due to the data set available. However, the primary message of low availability is clear (line 274).

• In contrast, the data presented on readiness is independent of delivery load. Perhaps it should be accordingly emphasized, as it is more meaningful. The increases described in 312-314 could also be emphasized.

• Tracer items: it is unclear whether the tracer items tracked have been selected from a larger list in the SARA manual, along with the justification for which items were included/not included.

• The extremely low percentage of Local HFs (0.3%) utilizing all 7 signal functions in the last 3 months may justify even more explanation in the discussion section.

• Grammar: further review needed for incomplete sentence (line 93), typos (line 415), increased use of articles (the, an, a, of), font change (391).

• The use of the term “availability” of signal functions, even if this is WHO standard, continues to be problematic. This was most obvious in lines 342-355, where the language suggested the presence of antibiotics, etc., instead of their utilization in the past 3 months (as described in lines 162-3). In other areas as well, the “availability” of signal functions implied a presence rather than utilization.

• Staff training in Table 2 is not defined until the discussion section and is still not clear. Does this refer to a provider receiving all the training courses in lines 390-392 in the last 2 years, or a portion, etc?

• Guidelines could also use clarification. Is this a standard set of government documents present in the facility?

• Partograph in Table 2 is also unclear. Is this the presence of a partograph in a patient record?

• Staff readjustment due to the pandemic could also be a factor in the loss of skill personnel in maternity wards.

• Figure 2 – Why is Local HF manual removal separate and not included in the third section?

• Overall, the figures are very helpful and instructive. Great additions.

Reviewer #2: Please find the following to be considered for revision.

1.In Abstract, Methods section-sentence no. 3, It would be better to use inferential statistics to cover mean, standard deviation, median and interquartile range…

2.In Abstract Section, Results-1st sentence, mention is Madesh and Gandaki are Facilities or Province or what is appropriate

3.The last sentence of the result section in abstract looks more like a conclusion, could the figures from your result table be added here instead

4.In introduction, 3rd paragraph, it is advisable to mention a full form for SMNH and EOC.

5.Likewise for PHCC, CHU, HP etc towards the last paragraph, please do mention their respective fullforms.

6.Introduction, last para, 3 rd line, “compared to no skill’ can be omitted

7.For Methodology section, is it possible to build a STROBE flow diagram to make things more clear at a glance

Reviewer #3: This revised paper on Basic Emergency Obstetric and Newborn Care in Nepal uses data from the National Health Facility Survey 2021. The authors have made substantial edits and have done well to respond to previous reviewer remarks. It is an important topic. The paper should be further strengthened as per the recommendations below.

1. The aim and objectives of the study are not clearly laid out. It just states that you aim to the service availability and readiness using the survey (line 93-97). This should be more specific to why you aim to do this assessment and what specific objectives you hope to achieve.

2. The trend analysis is an important component of the study and it features strongly in the discussion section. It should be more integrated throughout the paper. Currently, the first mention of 2015 is in the results (line 273), where is it embedded in the text and easily missed since there are many numbers presented in the two paragraphs where it is mentioned (273-275; 312-317). It is not included in the methods section or shown in any tables/figures in the paper (only the supplementary files). Given the focus of this part of the study in the discussion section, I suggest you also add is something in the introduction that justifies why you decided to look at the trends and what you would expect based on history (e.g. steady increase), add it as a specific objective of the study, explicitly mention the trend analysis in the methods and add to the related tables in the results.

3. Stillbirth prevention is a main outcome of EMO. Nearly the same number of stillbirths occur each year (1.9 million) as newborn deaths (2.4 million); half of stillbirths are intrapartum and preventable with access to high quality EMO. It is a major oversight to not include stillbirth throughout in your framing of EmONC as a way to reduce preventable mortality – furthering our argument that investing in EOC is a triple return on investment preventing maternal and newborn death and stillbirth. For more information, please see the UNICEF report: https://data.unicef.org/resources/never-forgotten-stillbirth-estimates-report/

4. The method used to calculate the readiness score of health facilities implies that each item has equal weight within each domain. In reality, we know that they are not equal in their efficacy to reduce mortality. This should be discussed as a limitation especially in lieu of your results.

5. The independent variables section (from line 176) is very text heavy and hard to read. I would suggest creating a table to show the variables rather than writing it out in lengthy text. A more detailed summary could be a supplementary file.

6. In the results section, you include too much text with specific numbers that duplicate what is in the table. It is hard to read and take away key findings. I suggest you tighten the results section text throughout and use it to show key examples or ranges.

7. Table 1 includes a key but the other tables do not. It would be helpful to add to all since it is already hard to digest the volume of numbers being shared in these tables.

8. You should make it clear what time point these results are for in tables 1-4. Given you have also looked at trends, it is not clear the year you have used for the data – this should be part of each title.

9. Grammar issues remain throughout. For example, line 94 is not a complete sentence. A full copy edit is needed.

10. The figures are blurry and suggest high resolution for publication.

11. The authors neglect to reference important and relevant studies from Nepal linked to EmONC and reflect how their study contributes to the literature. Some examples are listed below but this is not an exhaustive list. There is also a wealth of literature on EmONC including reviews as well as other studies (not referenced) that use SARA to assess EOC, which are not referenced and would enrich the discussion section.Without inclusion of how your study contributes to the literature, there remains the question of "so what?" especially to readers familiar with Nepal and use of large surveys (such as SARA and SPA) to assess EmONC.

References

Banstola, A., Simkhada, P., van Teijlingen, E., Bhatta, S., Lama, S., Adhikari, A., & Banstola, A. (2020). The Availability of Emergency Obstetric Care in Birthing Centres in Rural Nepal: A Cross-sectional Survey. Maternal and child health journal, 24(6), 806–816. https://doi.org/10.1007/s10995-019-02832-2

Chaulagain, D. R., Malqvist, M., Wrammert, J., Gurung, R., Brunell, O., Basnet, O., & Kc, A. (2022). Service readiness and availability of perinatal care in public hospitals - a multi-centric baseline study in Nepal. BMC pregnancy and childbirth, 22(1), 842. https://doi.org/10.1186/s12884-022-05121-z

Lewis, T. P., McConnell, M., Aryal, A., Irimu, G., Mehata, S., Mrisho, M., & Kruk, M. E. (2023). Health service quality in 2929 facilities in six low-income and middle-income countries: a positive deviance analysis. The Lancet. Global health, 11(6), e862–e870. https://doi.org/10.1016/S2214-109X(23)00163-8

Kc, A., Gurung, R., Kinney, M. V., Sunny, A. K., Moinuddin, M., Basnet, O., Paudel, P., Bhattarai, P., Subedi, K., Shrestha, M. P., Lawn, J. E., & Målqvist, M. (2020). Effect of the COVID-19 pandemic response on intrapartum care, stillbirth, and neonatal mortality outcomes in Nepal: a prospective observational study. The Lancet. Global health, 8(10), e1273–e1281. https://doi.org/10.1016/S2214-109X(20)30345-4

Sharma, G., Molla, Y. B., Budhathoki, S. S., Shibeshi, M., Tariku, A., Dhungana, A., Bajracharya, B., Mebrahtu, G. G., Adhikari, S., Jha, D., Mussema, Y., Bekele, A., & Khadka, N. (2021). Analysis of maternal and newborn training curricula and approaches to inform future trainings for routine care, basic and comprehensive emergency obstetric and newborn care in the low- and middle-income countries: Lessons from Ethiopia and Nepal. PloS one, 16(10), e0258624. https://doi.org/10.1371/journal.pone.0258624

7. PLOS authors have the option to publish the peer review history of their article (what does this mean?). If published, this will include your full peer review and any attached files.

Reviewer #1: **Yes: **Robert B. Clark, MD

Reviewer #2: No

Reviewer #3: No

---

## [Author Response · Author response to Decision Letter 1]

23 Jul 2023

Response to reviewers

Reviewer #1: Several minor points can use additional consideration:

Comment: Interpretation of the availability of signal functions is still problematic due to the mixture of low and high delivery loads at the Local HF. Further insight is limited due to the data set available. However, the primary message of low availability is clear (line 274).

Response: The data presented on readiness is independent of delivery load and we have tried to clarify the issue in discussion section.

Comment: Tracer items: it is unclear whether the tracer items tracked have been selected from a larger list in the SARA manual, along with the justification for which items were included/not included.

Response: Thank you for your feedback. The tracer items are selected based on SARA manual, previous literature and availability of data in NHFS 2021 and NHFS 2015.

Comment: The extremely low percentage of Local HFs (0.3%) utilizing all 7 signal functions in the last 3 months may justify even more explanation in the discussion section.

Response: Thank you. We have added explanation in the discussion section.

Comment: Grammar: further review needed for incomplete sentences (line 93), typos (line 415), increased use of articles (the, an, a, of), font change (391).

Response: Thank you for your feedback. We have revisited grammar and corrected wherever necessary as per your feedback.

Comment: The use of the term “availability” of signal functions, even if this is WHO standard, continues to be problematic. This was most obvious in lines 342-355, where the language suggested the presence of antibiotics, etc., instead of their utilization in the past 3 months (as described in lines 162-3). In other areas as well, the “availability” of signal functions implied a presence rather than utilization.

Response: Thank you for the feedback. NHFS takes into account the readiness and availability of the service rather than utilization. We have corrected sentences that give meaning of utilization in the manuscript.

Comment: Staff training in Table 2 is not defined until the discussion section and is still not clear. Does this refer to a provider receiving all the training courses in lines 390-392 in the last 2 years, or a portion, etc?

Response: Thank you for your comment. The definition of the tracer items, is explained in the supplementary table 1.

Comment: Guidelines could also use clarification. Is this a standard set of government documents present in the facility?

Response: Thank you for your comment. Yes, guidelines included in the survey are meant to assess the standard set of guidelines to be available in HFs. The definition of the tracer items, guidelines is kept in the supplementary table 1. 

Comment: Partograph in Table 2 is also unclear. Is this the presence of a partograph in a patient record?

Response: Thank you for your comment. The facility with blank partograph were considered to have partograph. The definition of the tracer items, partograph is placed in supplementary table 1.

Comment: Staff readjustment due to the pandemic could also be a factor in the loss of skill personnel in maternity wards.

Response: Thank you, we have included this explanation in discussion section.

Comment: Figure 2 – Why is Local HF manual removal separate and not included in the third section?

Response: Thank you for your comment. The error is fixed, and figure of highest resolution is upload in the portal

Comment: Overall, the figures are very helpful and instructive. Great additions.

Response: Thank you for your motivational words.

Response to comments from reviewer 2

Comment: In Abstract, Methods section-sentence no. 3, It would be better to use inferential statistics to cover mean, standard deviation, median and interquartile range…

Response: Thank you. We have revised the write up. But for the clarity of readers, we have also kept the list of descriptive statistics we used.

Comment: In Abstract Section, Results-1st sentence, mention is Madesh and Gandaki are Facilities or Province or what is appropriate

Response: It is already addressed in the manuscript.

Comment: The last sentence of the result section in abstract looks more like a conclusion, could the figures from your result table be added here instead

Response: It seems like this comment was applicable for our previous version of manuscript. We have updated the manuscript as suggested

Comment: In introduction, 3rd paragraph, it is advisable to mention a full form for SMNH and EOC.

Response: Thank you so much for the feedback. We have completely revised the manuscript. We have included full forms in first place they appear.

Comment: Likewise for PHCC, CHU, HP etc towards the last paragraph, please do mention their respective fullforms.

Response: Thank you. Comment has been addressed.

Comment: Introduction, last para, 3 rd line, “compared to no skill’ can be omitted

Response: Thank you. We have revised the manuscript as suggested.

Comment: For Methodology section, is it possible to build a STROBE flow diagram to make things more clear at a glance

Response: It is already addressed in the manuscript,

Response to comments from reviewer 3

Comment: The aim and objectives of the study are not clearly laid out. It just states that you aim to the service availability and readiness using the survey (line 93-97). This should be more specific to why you aim to do this assessment and what specific objectives you hope to achieve.

Response: We have tried to make the objectives more specific and included the reason for doing assessment.

Comment: The trend analysis is an important component of the study and it features strongly in the discussion section. It should be more integrated throughout the paper. Currently, the first mention of 2015 is in the results (line 273), where is it embedded in the text and easily missed since there are many numbers presented in the two paragraphs where it is mentioned (273-275; 312-317). It is not included in the methods section or shown in any tables/figures in the paper (only the supplementary files). Given the focus of this part of the study in the discussion section, I suggest you also add is something in the introduction that justifies why you decided to look at the trends and what you would expect based on history (e.g. steady increase), add it as a specific objective of the study, explicitly mention the trend analysis in the methods and add to the related tables in the results.

Response: Thank you for your insightful comment. Paper from 2015 analysis have been published and our analysis of NHFS 2021 is the first paper of its kind after Nepal moved to federal structure with major changes in the way health system is organized. In discussion section, we have compared our findings with previous publication from NHFS 2015. However, to prevent duplication with previous publications, we have confined our core analysis to NHFS 2021. We have revised contents throughout the manuscript to make it more clear and have cited source wherever there is comparison with 2015 findings in discussion section.

Comment: Stillbirth prevention is a main outcome of EMO. Nearly the same number of stillbirths occur each year (1.9 million) as newborn deaths (2.4 million); half of stillbirths are intrapartum and preventable with access to high quality EMO. It is a major oversight to not include stillbirth throughout in your framing of EmONC as a way to reduce preventable mortality – furthering our argument that investing in EOC is a triple return on investment preventing maternal and newborn death and stillbirth. For more information, please see the UNICEF report: https://data.unicef.org/resources/never-forgotten-stillbirth-estimates-report/

Response: Thank you for the feedback. We definitely agree that stillbirth prevention is one of the outcome of EMO. Our analysis is more focused on service availability rather than outcomes like NMR and stillbirths. 

Comment: The method used to calculate the readiness score of health facilities implies that each item has equal weight within each domain. In reality, we know that they are not equal in their efficacy to reduce mortality. This should be discussed as a limitation especially in lieu of your results.

Response: We agree that each items within the domains may not have equal efficacy. We believe that, determining weightage for each items included within domain based on their demonstrated efficacy and effectiveness in Nepalese context would be a separate study of high impact. We may strive in that direction in future. However, in this paper, we have taken reference from previous publications and SARA tool in computation of composite index and score.

Comment: The independent variables section (from line 176) is very text heavy and hard to read. I would suggest creating a table to show the variables rather than writing it out in lengthy text. A more detailed summary could be a supplementary file.

Response: Thank you for your comment. We have generated table showing definition of the variables as suggested.

Comment: In the results section, you include too much text with specific numbers that duplicate what is in the table. It is hard to read and take away key findings. I suggest you tighten the results section text throughout and use it to show key examples or ranges.

Response: Thank you for your insightful comments. We have reduced text in result section and include key findings in the result section

Comment: Table 1 includes a key but the other tables do not. It would be helpful to add to all since it is already hard to digest the volume of numbers being shared in these tables.

Response: Thank you for your comment, we have added key to all tables.

Comment: You should make it clear what time point these results are for in tables 1-4. Given you have also looked at trends, it is not clear the year you have used for the data – this should be part of each title.

Response: The NHFS dataset used in each tables are mentioned now as per your suggestion.

Comment: Grammar issues remain throughout. For example, line 94 is not a complete sentence. A full copy edit is needed.

Response: Thank you for your feedback. We have revisited grammar and corrected wherever necessary as per your feedback.

Comment: The figures are blurry and suggest high resolution for publication.

Response: Thank you for your comment. The figure of the highest resolution is upload in the portal but to our knowledge, figure are slightly blur when pdf was built. Based on our past experience, they will be clear when final version of paper appear in the website as original images will be used in the process. 

Comment: The authors neglect to reference important and relevant studies from Nepal linked to EmONC and reflect how their study contributes to the literature. Some examples are listed below but this is not an exhaustive list. There is also a wealth of literature on EmONC including reviews as well as other studies (not referenced) that use SARA to assess EOC, which are not referenced and would enrich the discussion section.Without inclusion of how your study contributes to the literature, there remains the question of "so what?" especially to readers familiar with Nepal and use of large surveys (such as SARA and SPA) to assess EmONC.

References

Banstola, A., Simkhada, P., van Teijlingen, E., Bhatta, S., Lama, S., Adhikari, A., & Banstola, A. (2020). The Availability of Emergency Obstetric Care in Birthing Centres in Rural Nepal: A Cross-sectional Survey. Maternal and child health journal, 24(6), 806–816. https://doi.org/10.1007/s10995-019-02832-2

Chaulagain, D. R., Malqvist, M., Wrammert, J., Gurung, R., Brunell, O., Basnet, O., & Kc, A. (2022). Service readiness and availability of perinatal care in public hospitals - a multi-centric baseline study in Nepal. BMC pregnancy and childbirth, 22(1), 842. https://doi.org/10.1186/s12884-022-05121-z

Lewis, T. P., McConnell, M., Aryal, A., Irimu, G., Mehata, S., Mrisho, M., & Kruk, M. E. (2023). Health service quality in 2929 facilities in six low-income and middle-income countries: a positive deviance analysis. The Lancet. Global health, 11(6), e862–e870. https://doi.org/10.1016/S2214-109X(23)00163-8

Kc, A., Gurung, R., Kinney, M. V., Sunny, A. K., Moinuddin, M., Basnet, O., Paudel, P., Bhattarai, P., Subedi, K., Shrestha, M. P., Lawn, J. E., & Målqvist, M. (2020). Effect of the COVID-19 pandemic response on intrapartum care, stillbirth, and neonatal mortality outcomes in Nepal: a prospective observational study. The Lancet. Global health, 8(10), e1273–e1281. https://doi.org/10.1016/S2214-109X(20)30345-4

Sharma, G., Molla, Y. B., Budhathoki, S. S., Shibeshi, M., Tariku, A., Dhungana, A., Bajracharya, B., Mebrahtu, G. G., Adhikari, S., Jha, D., Mussema, Y., Bekele, A., & Khadka, N. (2021). Analysis of maternal and newborn training curricula and approaches to inform future trainings for routine care, basic and comprehensive emergency obstetric and newborn care in the low- and middle-income countries: Lessons from Ethiopia and Nepal. PloS one, 16(10), e0258624. https://doi.org/10.1371/journal.pone.0258624

Response: Thank you for the feedback. This analysis is an analysis from nationally representative survey and we have sought to take reference of the article that are nationally representative with methodology comparable to NHFS 2021. In doing so, some article may have been missed. However, we further reviewed the article suggested and have cited relevant articles in our manuscript.

---

## [Editor Report · Decision Letter 2]

1 Aug 2023

Service Availability and Readiness for Basic Emergency Obstetric and Newborn Care: Analysis from Nepal Health Facility Survey 2021

PONE-D-23-04260R2

Dear Mr. Pandey

Thank you for revising the manuscript and responding to comments and recommendation.

We’re pleased to inform you that your manuscript has been judged scientifically suitable for publication and will be formally accepted for publication once it meets all outstanding technical requirements.

Kind regards,

Ashish KC, Associate Professor, University of Gothenburg, Sweden

Academic Editor

PLOS ONE
---

## [Editor Report · Acceptance letter]

8 Aug 2023

PONE-D-23-04260R2 

Service Availability and Readiness for Basic Emergency Obstetric and Newborn Care: Analysis from Nepal Health Facility Survey 2021 

Dear Dr. Pandey:

I'm pleased to inform you that your manuscript has been deemed suitable for publication in PLOS ONE. Congratulations! Your manuscript is now with our production department. 

Kind regards, 

on behalf of

Dr. Ashish KC 

Academic Editor

PLOS ONE